# Human Dectin-1 is *O*-glycosylated and serves as a ligand for C-type lectin receptor CLEC-2

**Shojiro Haji**[1,2†], **Taiki Ito**[1,2], **Carla Guenther**[1,2], **Miyako Nakano**[3], **Takashi Shimizu**[1,2], **Daiki Mori**[1,2‡, §], **Yasunori Chiba**[4], **Masato Tanaka**[5], **Sushil K Mishra**[6#], **Janet A Willment**[7], **Gordon D Brown**[7], **Masamichi Nagae**[1,2*], **Sho Yamasaki**[1,2,8,9*]

[1]Department of Molecular Immunology, Research Institute for Microbial Diseases, Osaka University, Osaka, Japan; [2]Laboratory of Molecular Immunology, Immunology Frontier Research Center (IFReC), Osaka University, Osaka, Japan; [3]Graduate School of Integrated Sciences for Life, Hiroshima University, Hiroshima, Japan; [4]Cellular and Molecular Biotechnology Research Institute, National Institute of Advanced Industrial Science and Technology (AIST), Tsukuba, Japan; [5]Laboratory of Immune Regulation School of Life Sciences, Tokyo University of Pharmacy and Life Sciences, Hachioji, Japan; [6]The Glycoscience Group, National University of Ireland, Galway, Galway, Ireland; [7]Medical Research Council Centre for Medical Mycology, University of Exeter, Exeter, United Kingdom; [8]Center for Infectious Disease Education and Research (CiDER), Osaka University, Osaka, Japan; [9]Division of Molecular Design, Research Center for Systems Immunology, Medical Institute of Bioregulation, Kyushu University, Fukuoka, Japan

**\*For correspondence:**
mnagae@biken.osaka-u.ac.jp (MN);
yamasaki@biken.osaka-u.ac.jp (SY)

**Present address:** [†]Department of Medicine and Bioregulatory Science, Graduate School of Medical Sciences, Kyushu University, Fukuoka, Japan; [‡]Centre d'Immunologie de Marseille-Luminy, Institut National de la Santé et de la Recherche Médicale, Centre National de laRecherche Scientifique, Marseille, France; [§]Centre d'Immunophénomique, AixMarseille Université, Institut National de la Santé et de la Recherche Médicale, Centre National de laRecherche Scientifique, Marseille, France; [#]Glycoscience Center of Research Excellence, The University of Mississippi, Mississippi, United States

**Competing interest:** The authors declare that no competing interests exist.

**Abstract** C-type lectin receptors (CLRs) elicit immune responses upon recognition of glycoconjugates present on pathogens and self-components. While Dectin-1 is the best-characterized CLR recognizing β-glucan on pathogens, the endogenous targets of Dectin-1 are not fully understood. Herein, we report that human Dectin-1 is a ligand for CLEC-2, another CLR expressed on platelets. Biochemical analyses revealed that Dectin-1 is a mucin-like protein as its stalk region is highly *O*-glycosylated. A sialylated core 1 glycan attached to the EDxxT motif of human Dectin-1, which is absent in mouse Dectin-1, provides a ligand moiety for CLEC-2. Strikingly, the expression of human Dectin-1 in mice rescued the lethality and lymphatic defect resulting from a deficiency of Podoplanin, a known CLEC-2 ligand. This finding is the first example of an innate immune receptor also functioning as a physiological ligand to regulate ontogeny upon glycosylation.

## Editor's evaluation

The C-type lectin receptor family recognises pathogens and self-components. Dectin-1 is known to recognize glucan on pathogens. In this fundamental study Dectin-1 and CLEC2 – another C-type lectin receptor expressed on platelets – interacts through an *O*-glycosylated ligand presented in the stalk region of Dectin-1. This compelling study demonstrates a potential role for pattern recognition receptors in physiological processes.

## Introduction

Innate immune receptors, mainly those characterized as pattern recognition receptors (PRRs), sense external stressors such as infectious agents. Upon recognizing pathogens, these PRRs trigger immune cell activation which leads to appropriate protective immune responses. PRRs also recognize self-derived endogenous components (*Marshak-Rothstein, 2006*), but the biological significance of this self-recognition is not fully understood.

C-type lectin receptors (CLRs), a family of PRRs, mainly signal via the immunoreceptor tyrosine-based activation motif (ITAM), which was originally identified in the cytoplasmic regions of the signaling subunits of antigen receptors which discriminate foreign from self-antigens. Accumulating evidence indicates that ITAM-coupled CLRs expressed on professional immune cells, such as macrophages and dendritic cells, play a critical role in protective immunity by recognizing various pathogens (*Iborra and Sancho, 2015*). ITAM-coupled CLRs are also expressed in non-professional immune cells; however, their beneficial roles are not fully understood.

Since CLRs recognize various types of ligands including glycans, glycolipids, lipids, proteins, and crystalline substances (*Brown et al., 2018*; *Iborra and Sancho, 2015*; *Zelensky and Gready, 2005*), individual target-focused omics approaches are limited in identifying their respective ligands. Instead, unbiased activity-based approaches have determined several individual targets (*Ahrens et al., 2012*; *Neumann et al., 2014*; *Yamasaki et al., 2008*).

Dectin-1, encoded by the *Clec7a* gene, is the first CLR identified as being ITAM-coupled (*Brown, 2006*). Its conserved carbohydrate recognition domain (CRD) mediates binding to fungal β-glucan, but the function(s) of other regions has not yet been determined. For example, the stalk region is missing in the short-splicing variant of Dectin-1 (Dectin-1B), which is expressed at a substantial level along with full-length Dectin-1 (Dectin-1A) (*Willment et al., 2001*; *Yokota et al., 2001*). This suggests that the stalk region might act as a functional unit; however, the function or even modification of the Dectin-1 stalk region is unknown.

Platelets are small anucleate blood cells generated from megakaryocytes in the bone marrow (*Battinelli et al., 2007*) that mediate pleiotropic functions including thrombosis or lymphangiogenesis. However, the roles of platelets beyond coagulation in the physiological processes are not fully understood (*Bertozzi et al., 2010*; *Osada et al., 2012*; *Ozaki et al., 2016*; *Suzuki-Inoue et al., 2017*; *Uhrin et al., 2010*). These processes are triggered by various receptors expressed on the surface of platelets (*Li et al., 2017*; *van der Meijden and Heemskerk, 2019*). Among them, CLEC-2, encoded by the *Clec1b* gene, is a Syk-coupled CLR expressed on platelets and megakaryocytes. During lymphangiogenesis, Podoplanin (Pdpn) expressed on lymphatic endothelial cells (*Suzuki-Inoue et al., 2007*) binds to CLEC-2 and induces platelet signaling via the hemITAM-Syk-SLP-76 axis, which is thought to regulate blood-lymph separation (*Abtahian et al., 2003*; *Bertozzi et al., 2010*).

Of note, platelet functions differ among mammalian species (*Sato and Harasaki, 2002*), implying that species-specific molecular interactions might regulate platelet function. However, to the best of our knowledge, no species-specific ligand(s) for platelet receptors have been identified, as human-specific machineries are the most difficult to study *in vivo*.

In the present study, using mass spectroscopic, biochemical, and genetic analyses, we identified human Dectin-1 as a glycosylated ligand for CLEC-2 providing a physiological role for PRR self-interaction.

## Results

### Human Dectin-1 interacts with CLEC-2 on platelets

To search for endogenous molecule(s) that interact with Dectin-1, we evaluated the ability of various organs to activate a Dectin-1-expressing reporter T cell line. Interestingly, human Dectin-1 (hDectin-1), but not mouse Dectin-1, reacted with several mouse organs, including liver, spleen, and bone marrow (*Figure 1A* and *Figure 1—figure supplement 1A*). T and B cells were dispensable for this interaction, as hDectin-1 binding was not decreased in splenocytes from Rag1-deficient mice (*Figure 1B*). Furthermore, splenocytes from mice with phenylhydrazine (PHZ)-induced anemia retained reactivity to hDectin-1 (*Figure 1C*), excluding erythrocytes as candidates. Upon density gradient separation of whole blood cells, the upper fraction interacted with hDectin-1, suggesting that platelets were the cell lineage expressing target(s) binding to hDectin-1 (*Figure 1D* and *Figure 1—figure supplement 1B*).

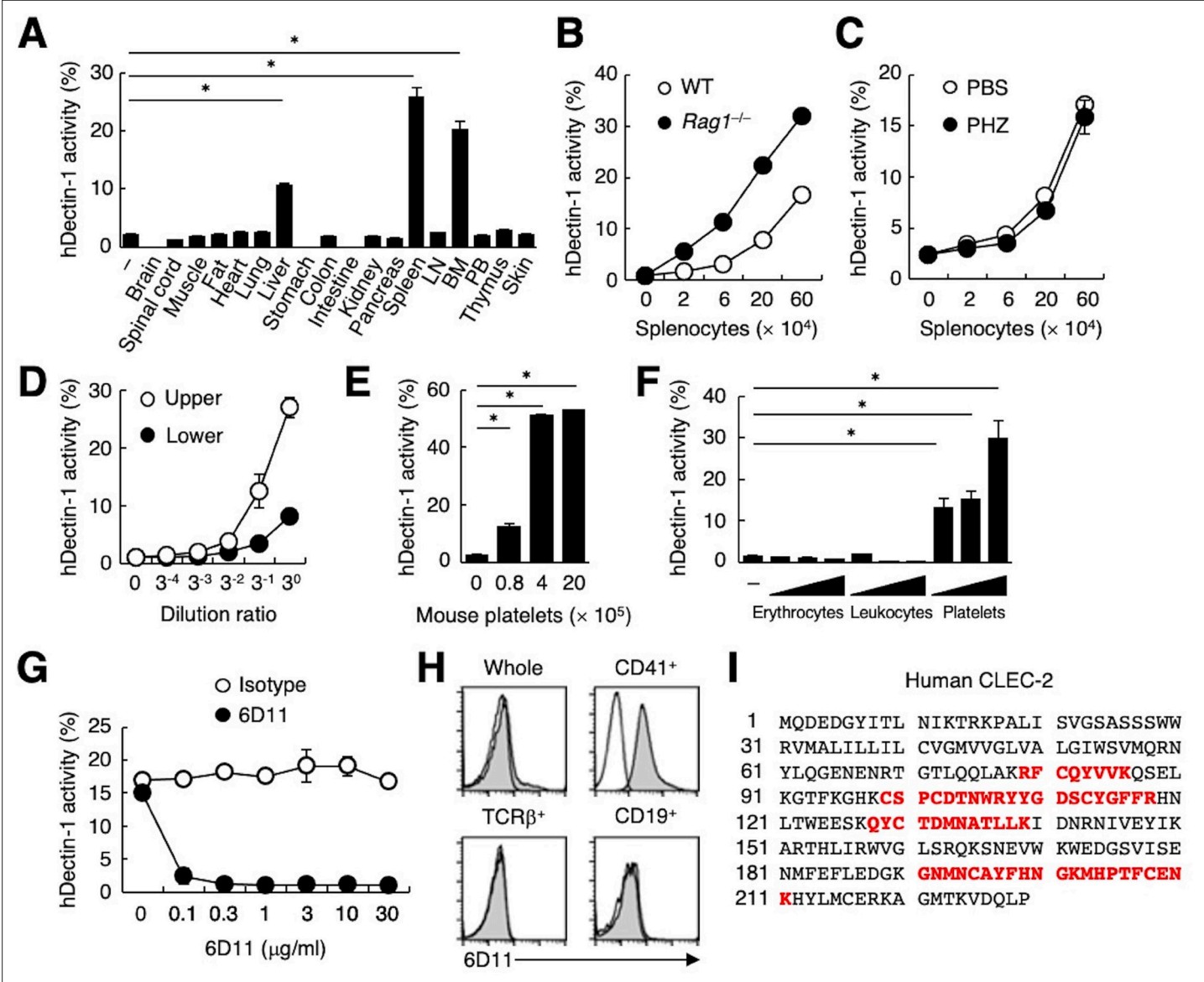

**Figure 1.** CLEC-2 expressed on platelets interacts with hDectin-1. (**A**) NFAT-GFP reporter cells expressing hDectin-1 were incubated with tissue homogenates derived from murine organs. LN: lymph node; BM: bone marrow; PB: peripheral blood; '−': cells without stimulants. (**B**) NFAT-GFP reporter cells expressing hDectin-1 were incubated with the indicated number of splenocytes from wild-type (WT, open circles) or Rag1-deficient mice (*Rag1−/−*, filled circles). (**C**) NFAT-GFP reporter cells expressing hDectin-1 were stimulated with the indicated number of splenocytes from mice treated with phosphate-buffered saline (PBS, open circles) or phenylhydrazine (PHZ, filled circles). (**D**) NFAT-GFP reporter cells expressing hDectin-1 were stimulated with the indicated dilution of upper (open circles) or lower (filled circles) fractions after separation by centrifugation of peripheral blood cells treated with Ammonium-Chloride-Potassium lysing buffer. (**E**) NFAT-GFP reporter cells expressing hDectin-1 were incubated with the indicated number of murine platelets. (**F**) NFAT-GFP reporter cells expressing hDectin-1 were stimulated with erythrocytes, leukocytes, or platelets (3, 10, and 30×10⁵ cells/well) isolated from human peripheral blood cells. '−': cells without stimulants. (**G**) NFAT-GFP reporter cells expressing hDectin-1 were incubated with human platelets in the absence or presence of the indicated concentration of an isotype control antibody (open circles) or 6D11 (filled circles). (**H**) 6D11-reactive cells were analyzed by flow cytometry in CD41⁺ platelets, TCR β⁺ T cells, and CD19⁺ B cells from peripheral blood cells using isotype control (rat IgG1 κ open histograms) or 6D11 (gray histograms). (**I**) Peptides in 6D11-immunoprecipitates detected by MALDI-TOF MS are highlighted in red within the entire amino acid sequence of hCLEC-2. The details of the detected peptides are shown in *Figure 1—figure supplement 1C*. Data in graphs are presented as mean ± SD. These results are representative of at least two independent experiments. An unpaired two-tailed Student t-test was used for all statistical analyses. *p<0.05.

The online version of this article includes the following source data and figure supplement(s) for figure 1:

**Figure supplement 1.** Interaction of human Dectin-1 with CLEC-2 on platelets.

**Figure supplement 1—source data 1.** Table for *Figure 1—figure supplement 1C*.

Indeed, purified mouse platelets bound to hDectin-1 (*Figure 1E*). Likewise, human platelets selectively interacted with hDectin-1, while erythrocytes and leukocytes did not (*Figure 1F*).

To identify molecules expressed on human platelets that interacted with hDectin-1, we raised rat monoclonal antibodies (mAbs) against human platelets. Among 1600 clones, only mAb clone 6D11 inhibited the binding between hDectin-1 and platelets (*Figure 1G*). In human peripheral blood cells, the 6D11 selectively reacted with CD41⁺ platelets (*Figure 1H*). We then used 6D11 to immunoprecipitate molecules from platelet lysates and identified CLEC-2 as the antigen recognized by 6D11 using mass spectrometry (MS) (*Figure 1I* and *Figure 1—figure supplement 1C*).

## Human Dectin-1 interacts with both human and mouse CLEC-2

The ectopic expression of human CLEC-2 (hCLEC-2) was sufficient to produce reactivity with 6D11 on a T cell hybridoma (*Figure 2A*). This transfectant acquired reactivity with hDectin-1 (*Figure 2B*), which was blocked by soluble 6D11 (*Figure 2C*).

Given that mouse platelets interacted with hDectin-1 (*Figure 1E*), we tested mouse CLEC-2 (mCLEC-2) and found that it interacted with Dectin-1 across species (*Figure 2D*). We next established CLEC-2-deficient mice (*Figure 2—figure supplement 1A and B*) and used fetal liver cells of these embryonic-lethal mice (*Bertozzi et al., 2010*) to reconstitute recipients thereby generating CLEC-2-deficient platelets. CLEC-2-deficient platelets lost the capacity to bind hDectin-1 (*Figure 2E and F*), indicating that mCLEC-2 on platelets is also essential for the 'xenogeneic' interaction with hDectin-1.

In contrast, and consistent with our initial screening (*Figure 1—figure supplement 1A*), mDectin-1 did not bind CLEC-2 on mouse or human platelets (*Figure 2—figure supplement 1C–E*). Thus, only hDectin-1 specifically interacted with CLEC-2 derived from various species.

Because CLEC-2 is a Syk-coupled activating receptor, we next tested whether hDectin-1 served as a ligand for CLEC-2. Ectopically expressed hDectin-1 activated CLEC-2 reporter cells (*Figure 2G*), suggesting that hDectin-1 signals through CLEC-2. Furthermore, co-culture of mouse platelets with hDectin-1 transfectants caused downregulation of CLEC-2 on the platelet cell surface, suggesting that hDectin-1 can engage CLEC-2 on platelets (*Figure 2H*). Indeed, peripheral CD14⁺ monocytes that express endogenous hDectin-1 activated hCLEC-2 reporter cells (*Figure 2I*).

## CLEC-2 CRD interacts with the hDectin-1, but not mDectin-1, stalk region

Because ligand activity was restricted to Dectin-1 of human origin, we assessed the key domain of Dectin-1 required for the interaction with CLEC-2 by designing domain-swapping mutants between human and mouse Dectin-1 (*Figure 3A*). Unexpectedly, the ligand-binding domain (CRD) of hDectin-1 was not required. Instead, the non-CRD region (a stretch including the cytoplasmic, transmembrane, and stalk regions) derived from hDectin-1 retained CLEC-2 ligand activity (*Figure 3A*). Importantly, these mutants retained the β-glucan receptor function (*Figure 3—figure supplement 1A*).

hDectin-1 has two major isoforms: the full length (1A) and a short isoform (1B) which lacks the stalk region (*Figure 3B*). Both isoforms acted as β-glucan receptors as previously reported (*Willment et al., 2001*; *Figure 3—figure supplement 1B*); however, only hDectin-1A bound CLEC-2, suggesting that the stalk region of hDectin-1 may mediate this unique ligand activity (*Figure 3B*). Consistent with this suggestion, a hDectin-1 truncation mutant that lacks the CRD but retains the stalk region (hDectin-1^ΔCRD) activated reporter cells that expressed CLEC-2 to an extent similar to full-length hDectin-1 (*Figure 3C*). Not surprisingly, hDectin-1^ΔCRD completely lost the β-glucan receptor function (*Figure 3—figure supplement 1C*).

In contrast, mutational analysis of hCLEC-2 revealed that a canonical ligand-binding domain (CRD) of hCLEC-2 was required for hDectin-1 binding activity (*Figure 3D*). Taken together, these results suggest that the hDectin-1 stalk region comprises a previously unappreciated function as a ligand recognized by the typical sugar-binding domain of the CLEC-2 receptor.

## CLEC-2 interacts with hDectin-1 harboring sialylated core 1 *O*-glycans

CLEC-2 recognizes its endogenous ligand Pdpn in an *O*-glycosylation-dependent manner (*Amano et al., 2008*; *Suzuki-Inoue et al., 2007*). Given that Dectin-1 has not been reported to be *O*-glycosylated, we examined the effect of glycosidases on Dectin-1. The apparent molecular mass of hDectin-1A in SDS-PAGE was larger than its estimated size, whereas *O*-glycosidase treatment produced

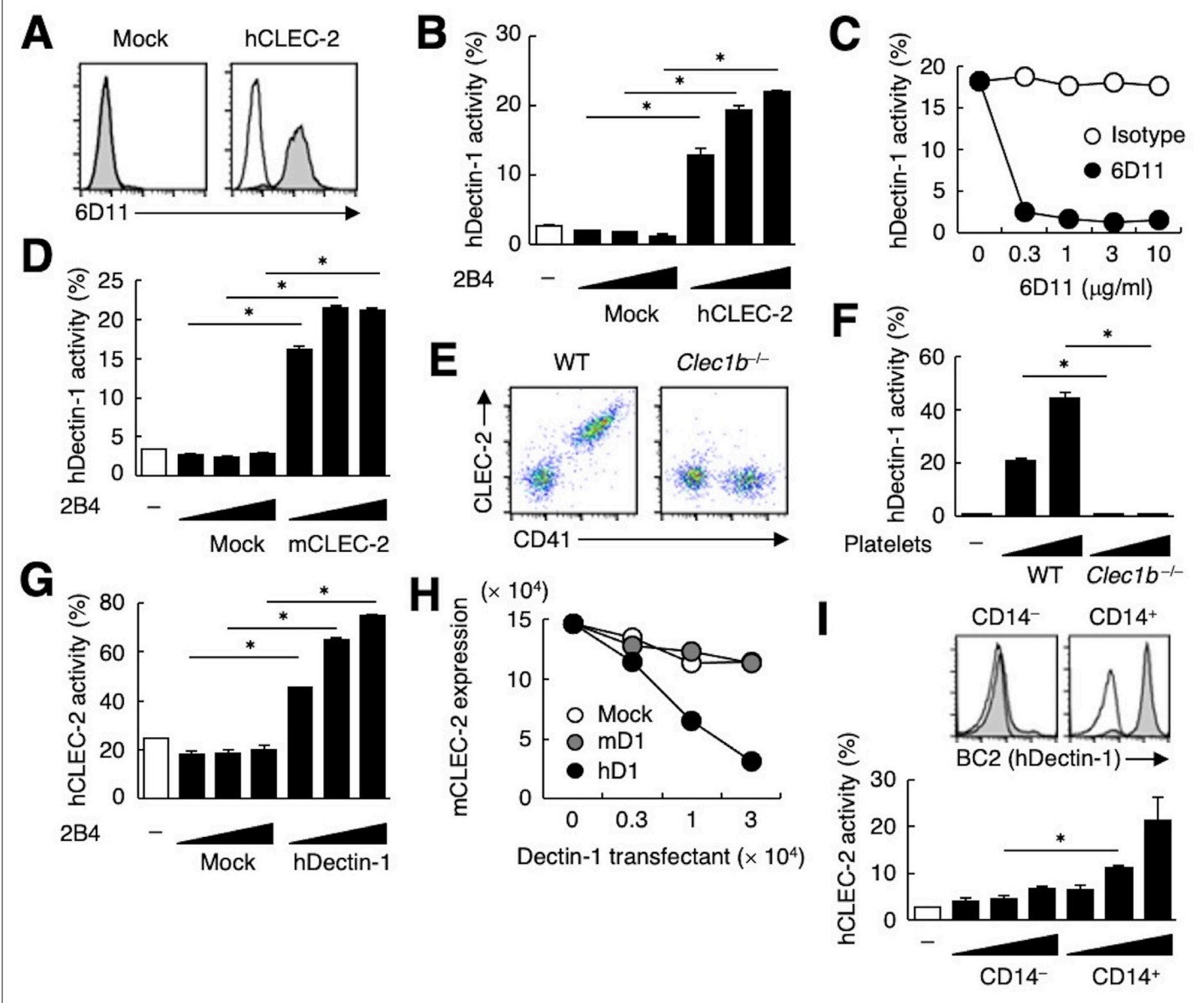

**Figure 2.** hDectin-1, but not mDectin-1, interacts with CLEC-2. (**A**) 2B4 cells expressing hCLEC-2/CD3 ζ or mock transfected cells were incubated with isotype control (rat IgG1 κ open histograms) or 6D11 (gray histograms). (**B**) NFAT-GFP reporter cells expressing hDectin-1 were incubated with control cells (mock transfectants) or 2B4 cells expressing hCLEC-2/CD3 ζ (0.3, 1, and 3×10⁴ cells/well). '−': cells without stimulants. (**C**) NFAT-GFP reporter cells expressing hDectin-1 were incubated with 2B4 cells expressing hCLEC-2/CD3 ζ in the absence or presence of the indicated concentrations of isotype control (rat IgG1 κ open circles) or 6D11 (filled circles). (**D**) NFAT-GFP reporter cells expressing hDectin-1 were incubated with mock transfectants or 2B4 cells expressing mouse CLEC-2/CD3 ζ (0.3, 1, and 3×10⁴ cells/well). '−': cells without stimulants. (**E**) Surface expression levels of CLEC-2 on platelets isolated from wild-type or *Clec1b⁻/⁻* mice were analyzed using flow cytometry. (**F**) NFAT-GFP reporter cells expressing hDectin-1 were incubated with platelets isolated from wild-type or *Clec1b⁻/⁻* mice (5 and 15×10⁵ cells/well). '−': cells without stimulants. (**G**) NFAT-GFP reporter cells expressing hCLEC-2 were incubated with mock transfectants or 2B4 cells expressing hDectin-1 (0.3, 1, and 3×10⁴ cells/well). '−': cells without stimulants. (**H**) Mouse platelets were incubated with the indicated number of 2B4 cells expressing mDectin-1 (mD1), or hDectin-1 (hD1) or mock transfectants. Surface expression levels of mCLEC-2 on platelets were subsequently analyzed using flow cytometry. Mean fluorescence intensity of anti-mCLEC-2 is indicated on the Y-axis as mCLEC-2 expression. Data are representative of two independent experiments. (**I**) CD14⁻ or CD14⁺ cells isolated from human peripheral blood cells were analyzed by flow cytometry using an isotype control (mouse IgG1 κ open histograms) or an anti-hDectin-1 antibody BC2 (gray histograms) (upper panel). NFAT-GFP reporter cells expressing hCLEC-2/CD3 ζ were incubated with CD14⁻ or CD14⁺ cells (1, 3, and 10×10⁴ cells/well) (lower panel). '−': cells without stimulants. Data are presented as mean ± SD. These results are representative of at least two independent experiments. An unpaired two-tailed Student t-test was used for all statistical analyses. *p<0.05.

The online version of this article includes the following figure supplement(s) for figure 2:

**Figure supplement 1.** Mouse and human CLEC-2 interact with hDectin-1.

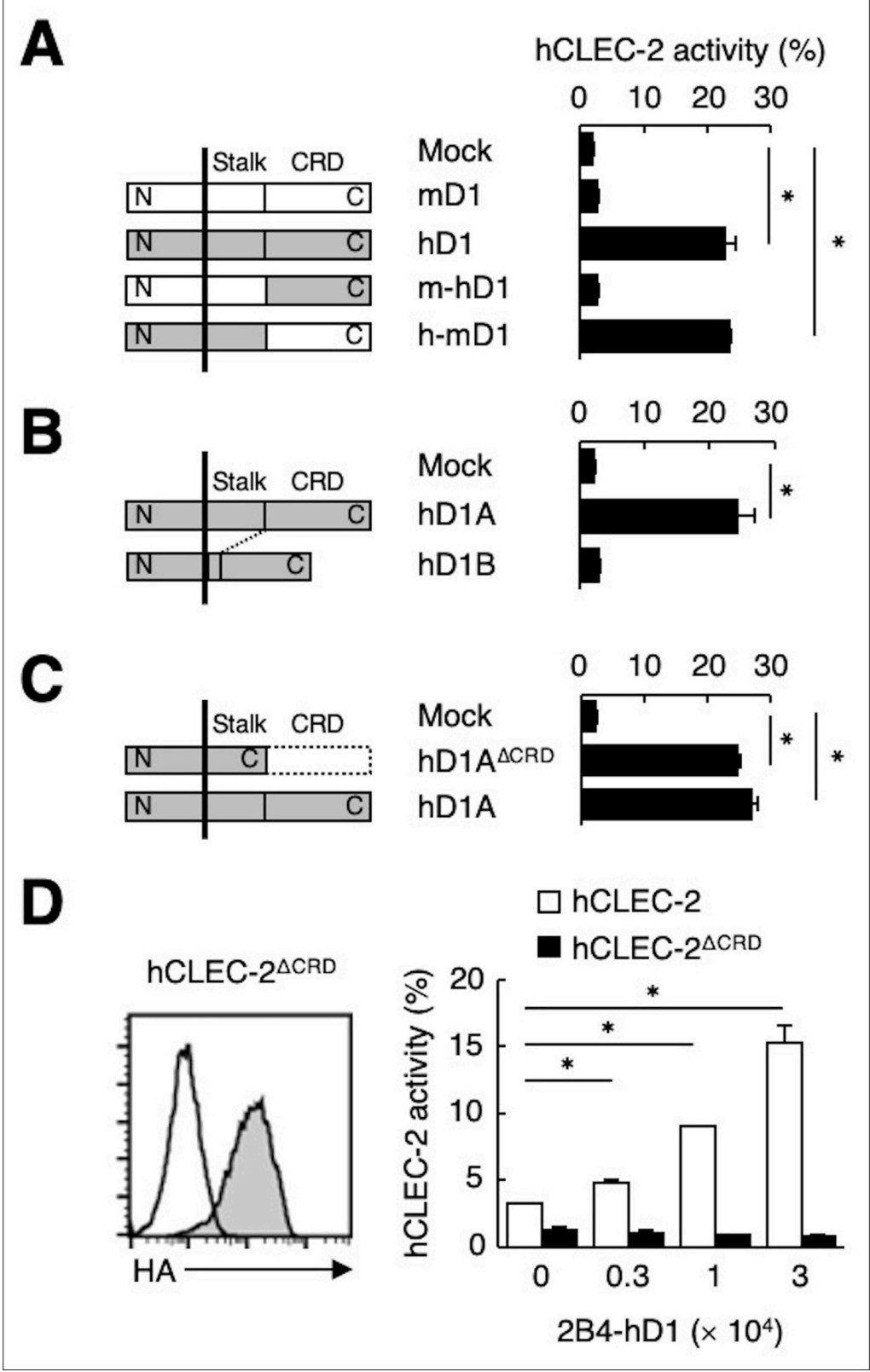

**Figure 3.** The stalk region of hDectin-1 interacts with the CRD of hCLEC-2. (**A**) Schematic representations of mDectin-1 (mD1), hDectin-1 (hD1), a mutant harboring mDectin-1 non-CRD and hDectin-1 CRD (m-hD1), and a mutant harboring hDectin-1 non-CRD and mDectin-1 CRD (h-mD1) are shown in the left panel. NFAT-GFP reporter cells expressing hCLEC-2/CD3ζ were incubated with NFAT-GFP reporter cells expressing mD1, hD1,

*Figure 3 continued on next page*

*Figure 3 continued*

m-hD1, h-mD1, or mock transfectants (right panel). Receptor activities of these Dectin-1 constructs are shown in *Figure 3—figure supplement 1A*. (**B**) Schematic representations of hDectin-1 isoforms A (hD1A) and B (hD1B) are shown in the left panel. NFAT-GFP reporter cells expressing hCLEC-2/CD3 ζ were incubated with NFAT-GFP reporter cells expressing hD1A, hD1B, or mock transfectants (right panel). Receptor activities of hD1A and hD1B are shown in *Figure 3—figure supplement 1B*. (**C**) Schematic representations of a hDectin-1A mutant lacking the CRD (hD1A^ΔCRD) and full-length (hD1A) are shown in the left panel. NFAT-GFP reporter cells expressing hCLEC-2/CD3 ζ were incubated with NFAT-GFP reporter cells expressing hD1A^ΔCRD, hD1A, or mock transfectants (right panel). Receptor activities of hD1A and hD1A^ΔCRD are shown in *Figure 3—figure supplement 1C*. (**D**) Expression levels of an HA-tagged hCLEC-2/CD3 ζ mutant lacking the CRD (hCLEC-2^ΔCRD) in NFAT-GFP cells was assessed by flow cytometry using an anti-HA tag antibody (gray histograms) (left panel). Open histogram represents unstained controls. NFAT-GFP reporter cells expressing hCLEC-2/CD3 ζ or hCLEC-2^ΔCRD were incubated with 2B4 cells expressing hDectin-1 (2B4-hD1) (right panel). Data are presented as mean ± SD. These results are representative of three independent experiments. An unpaired two-tailed Student t-test was used for all statistical analyses. *p<0.05.

The online version of this article includes the following figure supplement(s) for figure 3:

**Figure supplement 1.** Glycosylation of the stalk region of hDectin-1 mediates recognition by CLEC-2.

a significant downshift by approximately 10 kDa (*Figure 4A*). In contrast, *O*-glycosidase treatment did not affect hDectin-1B, the hDectin-1 isoform lacking a stalk, implying that hDectin-1A is *O*-glycosylated at the stalk region.

We also addressed this possibility using the selective endoprotease, OpeRATOR, which specifically cleaves at *O*-glycosylated Ser/Thr residues harboring β1–3 linked disaccharide units. hDectin-1A was cleaved by this protease, whereas hDectin-1B was not (*Figure 4B*), confirming that the stalk region of hDectin-1 is *O*-glycosylated. The substrate specificity of OpeRATOR suggests that hDectin-1A possesses core 1 (Galβ1-3GalNAc) or core 3 (GlcNAcβ1-3GalNAc) *O*-glycans within its stalk region (*Figure 4C*).

To determine the glycan structure on hDectin-1 required for the CLEC-2 interaction, we genetically modified the requisite biosynthesizing enzymes using CRISPR/Cas9. Disruption of the C1GalT1-specific chaperone (COSMC) in hDectin-1-expressing cells eliminated the capacity of hDectin-1 to bind CLEC-2 as well as PNA and MAL-II lectins, which recognize core 1 and sialylated disaccharide units, respectively. The re-expression of COSMC rescued these interactions (*Figure 4D*), suggesting that core 1 *O*-glycans (Galβ1-3GalNAc) are required for the hDectin-1−CLEC-2 interaction.

In contrast, and consistent with the specificity of OpeRATOR, core 2 *O*-glycans (Galβ1-3[GlcNAcβ1–6] GalNAc) were not necessary for CLEC-2 binding, as forced addition of GlcNAc onto core 1 by enzyme overexpression (core 2 β1,6 *N*-acetylglucosaminyltransferase, C2GnT) (*Figure 4C*) reduced CLEC-2-mediated reporter activity (*Figure 4E*). The requirement for terminal sialic acid was further confirmed by treatment with sialidase (*Figure 4F*). Since the surface expression levels of hDectin-1 were not affected throughout these modifications (*Figure 3—figure supplement 1D and E*), these results suggest that sialylated core 1 *O*-glycans on the hDectin-1 stalk region are key to CLEC-2 interaction.

## Two acidic residues and the *O*-glycan on T105 are required for hDectin-1 binding of CLEC-2

To further verify the requirement of *O*-glycosylation on the stalk region of hDectin-1 for binding of CLEC-2, we mutated all 14 Ser/Thr residues within the stalk region to Ala (hDectin-1.14A) (*Figure 5—figure supplement 1A*). hDectin-1.14A lost CLEC-2 ligand activity, but retained β-glucan receptor function (*Figure 5A*, and *Figure 5—figure supplement 1B and C*). Thus, either of these Ser or Thr residues is a critical site for *O*-glycosylation that mediates CLEC-2 binding.

We next analyzed the structure of the *O*-glycan attached to hDectin-1 using liquid chromatography-electrospray ionization (LC-ESI) MS analysis of digested peptide fragments of immunoprecipitated hDectin-1. Given the size of the fragments produced by cleavage with OpeRATOR (*Figure 4B*), Thr residues within the stalk region (T105, T107, T112, and T113) are predicted to be *O*-glycosylation sites (*Steentoft et al., 2013*). Mass spectrometric analysis detected peptide fragments showing additional *m/z* shifts (SVTPTK), which include T105 and T107. Among the possible *O*-glycans deduced from their *m/z* values (*Figure 5—figure supplement 2*), the glycan structures that satisfied the above-mentioned criteria were mono- or disialyl-core 1 *O*-glycans attached to T105 or T107 (*Figure 5B*).

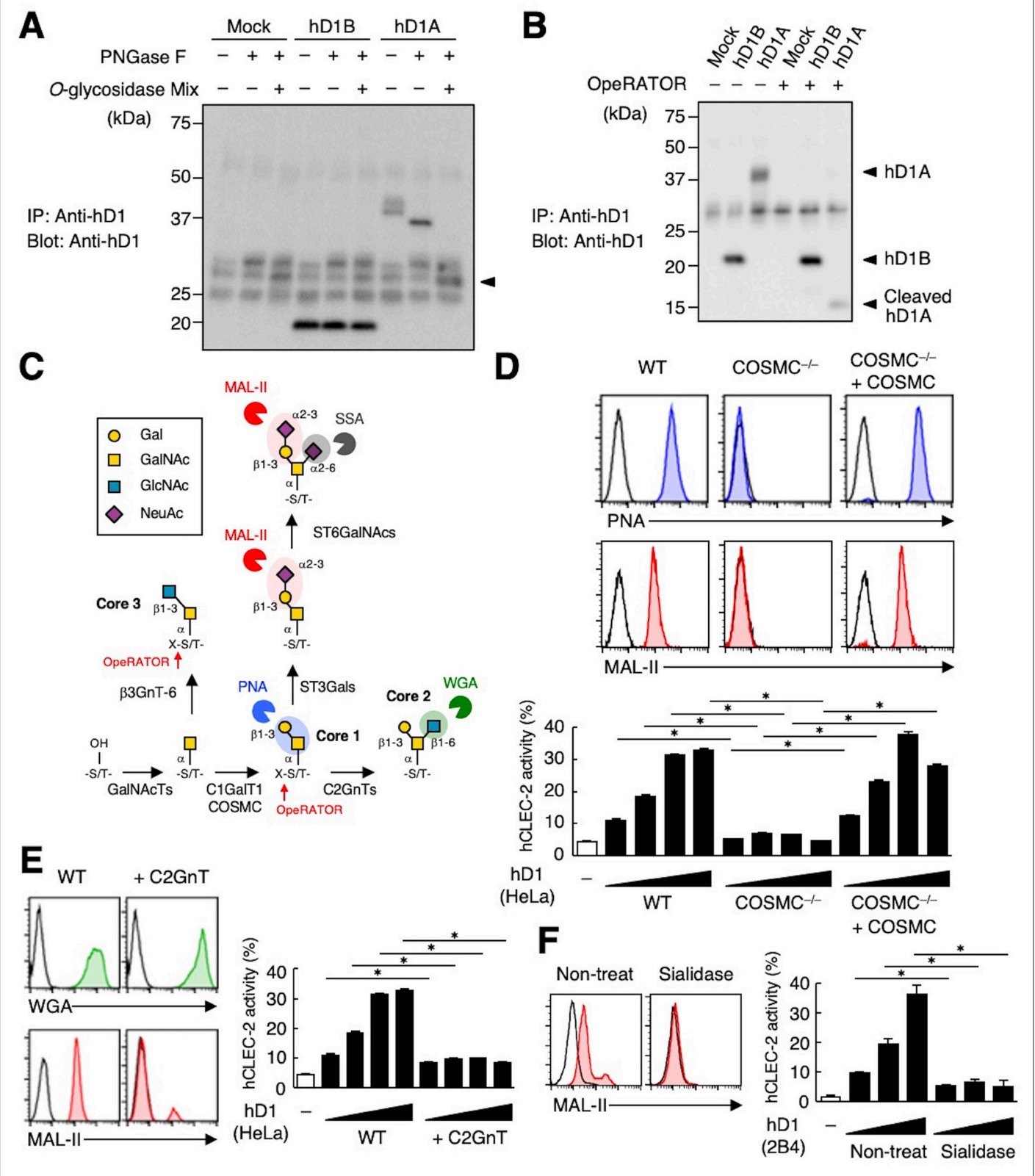

**Figure 4.** The sialylated core 1 *O*-glycan of hDectin-1 is required for its interaction with CLEC-2. (**A**) Mock transfectants, and hDectin-1B (hD1B) and hDectin-1A (hD1A) were immunoprecipitated with anti-hDectin-1 antibody and treated with PNGase F alone or PNGase F with *O*-glycosidase mix which contains *O*-glycosidase, Neuraminidase and *N*-acetylhexosaminidase. The calculated molecular mass of the core hDectin-1A polypeptide (27 kDa) is indicated with an arrowhead. '−': cells without indicated glycosidases. The raw images of Western blot were shown in ***Figure 4—source***

*Figure 4 continued on next page*

*Figure 4 continued*

**data 1**. (**B**) Mock transfectants and hDectin-1B (hD1B) and hDectin-1A (hD1A) transfectants were immunoprecipitated with anti-hDectin-1 antibody and treated with the glycosylation-specific protease OpeRATOR. Intact and cleaved hDectin-1 were detected using an anti-hDectin-1 polyclonal antibody (arrowheads). '−': cells without OpeRATOR. The raw images of Western blot were shown in **Figure 4—source data 2**. (**C**) Schematic drawing of the *O*-glycosylation pathway modified according to **Varki, 2017**. Monosaccharides are depicted according to the Symbol Nomenclature for Graphical Representation of Glycans (**Varki et al., 2015**). The responsible glycosyltransferases in each pathway are labeled adjacent to arrows. The abbreviations of these enzymes are as following; GalNAcT: *N*-acetylgalactosaminyltransferase, C1GalT1: core 1 β1–3 galactosyltransferase 1, COSMC: C1GalT1-specific chaperone, C2GnT: core 2 β1–6 *N*-acetylglucosaminyltransferase, β3GnT6: β1–3 *N*-acetylglucosaminyltransferase 6, ST3Gal: *N*-acetyllactosaminide α2–3 sialyltransferase, ST6GalNAc: *N*-acetylgalactosamine α2–6 sialyltransferase. The ligand specificities of the MAL-II, PNA, WGA, and SSA lectins are also highlighted in red, blue, green, and gray, respectively. The cleavage sites of OpeRATOR in core 1 and 3 glycans are indicated by red arrows. (**D**) Surface expression levels of core 1 and NeuAcα2–3 Gal disaccharide units on wild-type (WT), COSMC knockout (COSMC$^{-/-}$), and recaptured COSMC (COSMC$^{-/-}$+COSMC) were detected using PNA (blue histograms) or MAL-II lectins (red histograms) (upper panels). NFAT-GFP reporter cells expressing hCLEC-2/CD3 ζ were incubated with wild-type, COSMC knockout, and recaptured COSMC HeLa cells (0.1, 0.3, 1, and 3×10$^4$ cells/well) (lower panel). '−': cells without stimulants. Surface expression levels of hDectin-1 in wild-type or mutated cells are shown in **Figure 3—figure supplement 1D**. (**E**) The surface expression levels of GlcNAc and NeuAcα2–3 Gal on wild-type and C2GnT-overexpressing HeLa cells were detected using WGA (green histograms) and MAL-II lectin (red histograms) (left panels). NFAT-GFP reporter cells expressing hCLEC-2/CD3 ζ were incubated with wild-type (WT) and C2GnT-overexpressing (+C2 GnT) hDectin-1 (hD1) transfectant HeLa cells (0.1, 0.3, 1, and 3×10$^4$ cells/well) (right panel). '−': cells without stimulants. Surface expression levels of hDectin-1 in wild-type and C2GnT-overexpressing cells are shown in **Figure 3—figure supplement 1D**. (**F**) The surface expression levels of NeuAcα2–3 Gal on untreated and sialidase-treated 2B4 cells expressing hDectin-1 were detected using MAL-II lectins (red histograms) (left panels). NFAT-GFP reporter cells expressing hCLEC-2/CD3 ζ were incubated with untreated and sialidase-treated 2B4 cells expressing hDectin-1 (hD1) (0.3, 1, and 3×10$^4$ cells/well) (right panel). '−': cells without stimulants. Surface expression levels and receptor activities of hDectin-1 in non-treated and sialidase-treated cells are shown in **Figure 3—figure supplement 1E**. Data are presented as mean ± SD. These results are representative of at least three independent experiments. An unpaired two-tailed Student t-test was used for all statistical analyses. *p<0.05.

The online version of this article includes the following source data for figure 4:

**Source data 1.** Unedited and labeled Western blot in **Figure 4A**.

**Source data 2.** Unedited and labeled Western blot in **Figure 4B**.

By introducing individual substitutions at T105 and T107, we found that hDectin-1$^{T105A}$, but not hDectin-1$^{T107A}$, affected the ligand activity (**Figure 5C**). Consistent with this, the restoration of T105 into the hDectin-1.14A mutant, which was confirmed to be glycosylated in this context (**Figure 5—figure supplement 1D**), was sufficient to activate CLEC-2 (**Figure 5D**). These data suggest that the α2–3-linked monosialyl- or α2–3- and α2–6-linked disialyl-core 1 *O*-glycan at T105 of hDectin-1 is the key characteristic determining CLEC-2 ligand activity (MonoSia T$^{105}$ or DiSia T$^{105}$ in **Figure 5B**), as α2–6-linked monosialyl-core 1, sialyl-Tn, is rarely observed in normal tissue (**Pinho and Reis, 2015**). Further addition of sialic acids on core 1 by overexpression of ST6GalNAc4 (**Figure 4C**) slightly increased both DiSia-specific probe binding and CLEC-2 reporter activity (**Figure 5E**), suggesting that the disialylated core 1 is the preferred active determinant on T105 of hDectin-1 (DiSia T$^{105}$ in **Figure 5B**). We therefore synthesized hDectin-1 peptides harboring this *O*-glycan at T105. As expected, *O*-glycosylated peptides, but not non-glycosylated peptides, interacted with immobilized CLEC-2 CRD in surface plasmon resonance (SPR) assays (**Figure 5F**). These results suggest SSLEDSVT$^{105}$PTK attached with the disialylated core 1 *O*-glycan at T105 as a key ligand moiety.

## Comparison of hDectin-1 and Pdpn as a CLEC-2 ligand

We noticed that the glycan and amino acid motifs in hDectin-1 (EDSVT) are similar to those of Pdpn (EDDVVT) (**Kaneko et al., 2007**; **Figure 5—figure supplement 1E**). However, this EDxxT motif and CLEC-2 ligand activity are observed only in Dectin-1 from higher primates, but not mice (**Figure 5G**), which is in sharp contrast to the Pdpn EDxxxT motif conserved across various species (**Figure 5H**). Thus, Dectin-1 is a species-specific ligand for CLEC-2, albeit the β-glucan-binding CRD is well-conserved (**Figure 5—figure supplement 1F and G**).

Molecular modeling of the complex structure of hDectin-1 and CLEC-2 provided a suggested contribution of E101 and D102 as well as the T105-attached glycan to the interaction (**Figure 5I**). Indeed, E101 and D102 of hDectin-1 were essential for the binding to CLEC-2 (**Figure 5J** and **Figure 5—figure supplement 1B–C**). As for receptor reactivity, four Arg residues within the CLEC-2 CRD were critical for hDectin-1 (**Figure 5K**) as well as Pdpn recognition (**Nagae et al., 2014**). Thus, hDectin-1 is recognized by the CLEC-2 CRD which also mediates Pdpn recognition in a similar mode

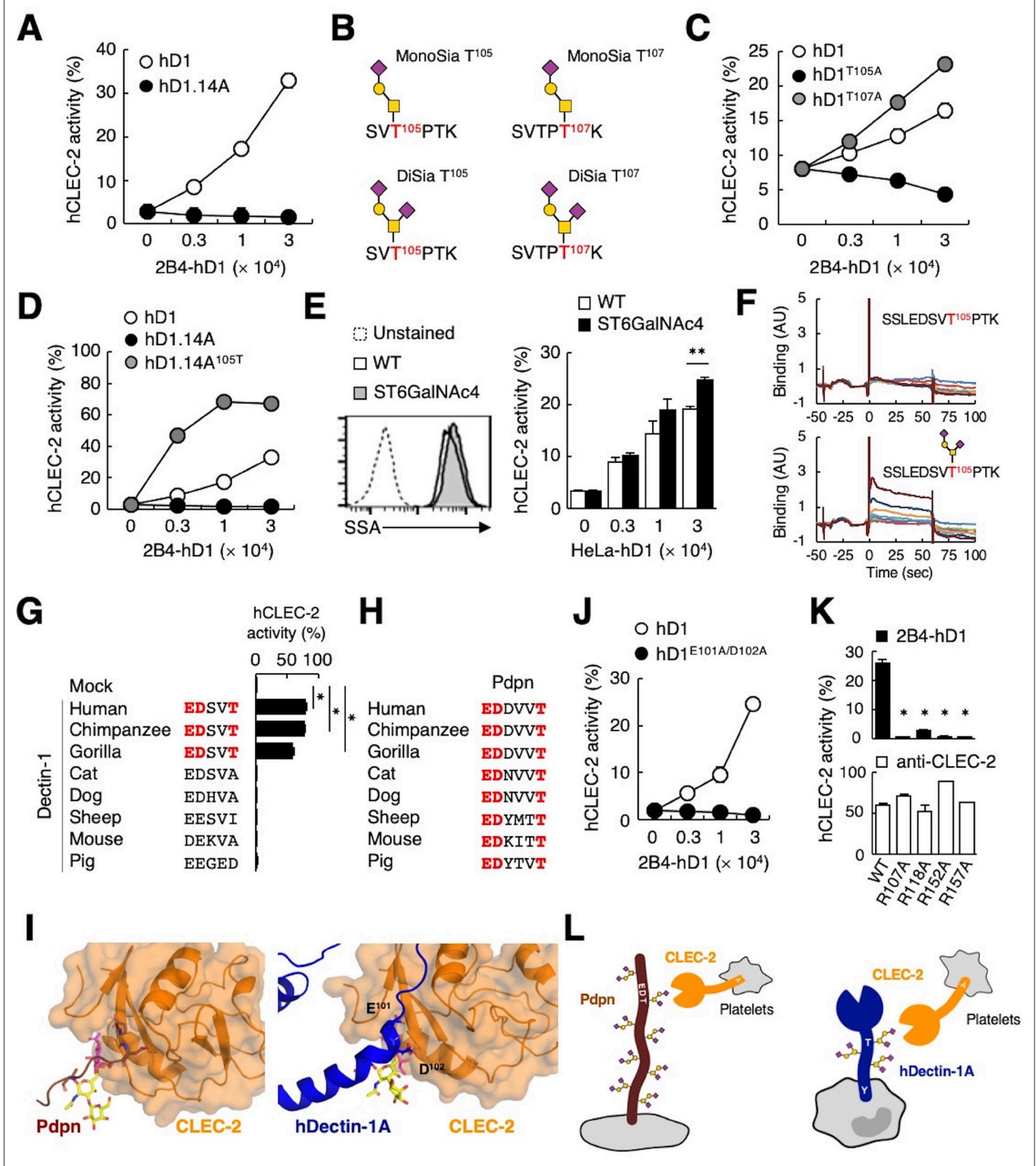

**Figure 5.** The sialylated *O*-glycan at T105 and adjacent amino acids are required for hDectin-1 binding of CLEC-2. (**A**) Interaction of CLEC-2 with wild-type or an *O*-glycan-less mutant of hDectin-1 (hD1.14A). NFAT-GFP reporter cells expressing hCLEC-2/CD3 ζ were incubated with the indicated number of 2B4 cells expressing wild-type hDectin-1 (hD1, open circles) and hD1.14A (filled circles). Details of the mutated constructs are summarized in ***Figure 5—figure supplement 1A***. (**B**) *O*-glycan structures attached at T105 or T107 as detected by LC-ESI MS analysis. A monosialylated (MonoSia)

*Figure 5 continued on next page*

*Figure 5 continued*

or disialylated (DiSia) glycan was potentially attached at T105 or T107. Details of the mass spectrometric analysis are shown in *Figure 5—figure supplement 2*. (**C**) Effect of alanine mutations in T105 or T107. NFAT-GFP reporter cells expressing hCLEC-2/CD3 ζ were incubated with the indicated number of 2B4 cells expressing wild-type (open circles), T105A (hD1$^{T105A}$, black circles), and T107A mutants of hDectin-1 (hD1$^{T107A}$, gray circles). Details of the mutated constructs are summarized in *Figure 5—figure supplement 1A*. (**D**) Effect of reconstitution of T105 within the hD1.14A-expressing mutant on hDectin-1−CLEC-2 interactions. NFAT-GFP reporter cells expressing hCLEC-2/CD3 ζ were incubated with the indicated cell number of 2B4 cells expressing wild-type (open circles), hD1.14A (black circles), or hD1.14A mutant reconstituted with T105 (hD1.14A$^{105T}$) (gray circles). Details of the mutated constructs are summarized in *Figure 5—figure supplement 1A*. (**E**) Effect on the hDectin-1−CLEC-2 interaction of overexpression of ST6GalNAc4 using HeLa cells expressing hDectin-1. The surface expression levels of α2–6-linked sialic acid on wild-type (open histograms) and ST6GalNAc4-overexpressing HeLa cells (gray histograms) were detected using SSA lectin (left panel). Dotted line represents unstained control. NFAT-GFP reporter cells expressing hCLEC-2/CD3 ζ were incubated with the indicated number of wild-type and ST6GalNAc4-overexpressing HeLa cells (right panel). (**F**) Surface plasmon resonance assay of recombinant CLEC-2 CRD and non-glycosylated (upper panel) or fully glycosylated (lower panel) hDectin-1 peptides. (**G**) Multiple amino acid sequence alignment of Dectin-1 derived from various mammalian species (left panel). The critical amino acid residues for CLEC-2 interaction are highlighted in red. NFAT-GFP reporter cells expressing hCLEC-2/CD3 ζ were incubated with NFAT-GFP reporter cells expressing Dectin-1 derived from various species, or mock transfectants (right panel). (**H**) Multiple amino acid sequence alignment of Pdpn derived from various mammalian species. The critical amino acid residues for CLEC-2 interaction are highlighted in red. (**I**) Structural comparison between crystal structure of the CLEC-2 CRD−Pdpn complex (PDB code: 3WSR) (left panel) and docking model of the CLEC-2 CRD−hDectin-1 complex (right panel). These two figures are depicted from the same angle of view. (**J**) Effect of the E101A/D102A hDectin-1 mutant (hD1$^{E101A/D102A}$) on the interaction with hCLEC-2. NFAT-GFP reporter cells expressing hCLEC-2/CD3 ζ were incubated with the indicated number of 2B4 cells expressing hD1 (open circles) or hD1$^{E101A/D102A}$ (filled circles). Details of the mutated constructs are summarized in *Figure 5—figure supplement 1A*. (**K**) Effect of hCLEC-2 mutants on interactions with hDectin-1 or an anti-CLEC-2 antibody. Four Arg residues (R107, R118, R152, and R157) of hCLEC-2, which are required for the interaction with Pdpn, were replaced with Ala residues. NFAT-GFP reporter cells expressing wild-type or mutated hCLEC-2/CD3 ζ were co-cultured with 2B4 cells expressing hDectin-1 (hD1, upper panel) or stimulated with plate-coated anti-hCLEC-2 antibody (lower panel). Data are presented as mean ± SD. *p<0.05, versus WT. (**L**) Schematic drawings of the CLEC-2−Pdpn interaction and the CLEC-2−hDectin-1 interaction. These results are representative of at least two independent experiments. An unpaired two-tailed Student t-test was used for all statistical analyses. **p<0.01, *p<0.05.

The online version of this article includes the following source data and figure supplement(s) for figure 5:

**Figure supplement 1.** Glycosylation motif in hDectin-1 required for CLEC-2 binding.

**Figure supplement 1—source data 1.** Unedited and labeled Western blot in *Figure 5—figure supplement 1D*.

**Figure supplement 2.** *O*-Glycan-structural analysis of hDectin-1 using LC-ESI MS.

(*Figure 5L*). In contrast to the restricted expression of Pdpn in lymphatic endothelial cells, hDectin-1 is widely expressed (*Willment et al., 2005*), implying that hDectin-1 might have functions as a ligand for CLEC-2 that differ from Pdpn.

We therefore compared the potency of hDectin-1 and Pdpn as CLEC-2 ligands by coculturing transfectants expressing hDectin-1 or Pdpn with CLEC-2 reporter cells. hDectin-1 induced CLEC-2 reporter activity less efficiently than Pdpn (*Figure 6A*). These different sensitivities may generate drastic qualitative changes in cellular responses, as Pdpn induced vigorous platelet aggregation, whereas hDectin-1 did not (*Figure 6B*). These results suggest that hDectin-1 is a weak CLEC-2 ligand that does not reach the threshold of platelet aggregation.

## Dectin-1 expression rescues the blood-lymph mixing phenotype of podoplanin-deficient mice

Pdpn is indispensable as a CLEC-2 ligand for lymphangiogenesis in mice (*Bertozzi et al., 2010*; *Schacht et al., 2003*). To evaluate whether hDectin-1 compensates for Pdpn deficiency as an alternative CLEC-2 ligand, we established hDectin-1 transgenic (Tg) mice and confirmed its expression and glycosylation in mice (*Figure 6C and D*). We also generated Pdpn-deficient mice (*Figure 6—figure supplement 1A and B*) and crossed them with hDectin-1 Tg mice. Although, as expected, no *Pdpn*$^{-/-}$ pups were born (*Ramirez et al., 2003*), hDectin-1 Tg×*Pdpn*$^{-/-}$ pups were obtained (*Table 1*).

To examine whether hDectin-1 mediated embryonic angiogenesis, we further surveyed blood and lymphatic vessels. Compared with wild-type (WT) mice, *Pdpn*$^{-/-}$ mice exhibited edema and blood-perfused lymphatic vessels in the skin on the back as reported previously (upper and middle panels in *Figure 6E*; *Uhrin et al., 2010*). The lymphatic vessels of *Pdpn*$^{-/-}$ mice were dilated, rugged, and filled with red blood cells (*Figure 6F and G*), confirming that blood-lymph separation was impaired (*Suzuki-Inoue et al., 2010*). However, in hDectin-1 Tg×*Pdpn*$^{-/-}$ embryos, red blood cells were excluded from straight and smooth lymphatic vessels and only detected in blood vessels (bottom panels in *Figure 6F*

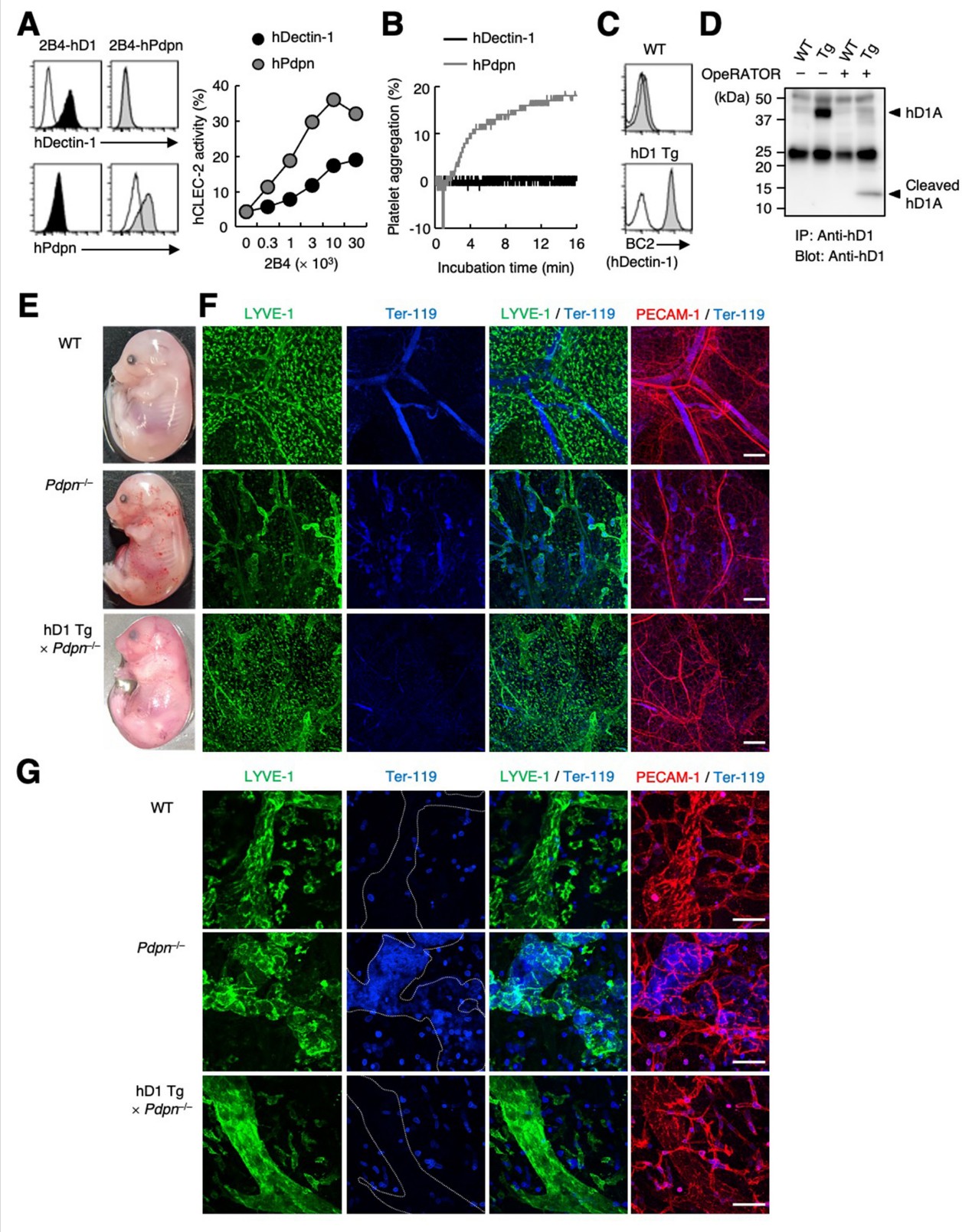

**Figure 6.** hDectin-1 does not induce platelet aggregation but rescues embryonic lethality in Pdpn-null mice. (**A**) The surface expression levels of hDectin-1 (black histograms) and human Pdpn (hPdpn, gray histograms) in 2B4 transfectant cells were evaluated with anti-hDectin-1 and anti-hPdpn antibodies, respectively (left panels). Open histograms represent unstained controls. NFAT-GFP reporter cells expressing hCLEC-2/CD3ζ were stimulated with the indicated cell number of 2B4 cells expressing hDectin-1 (black circles) or hPdpn (gray circles) (right panel). Data are presented as

*Figure 6 continued on next page*

*Figure 6 continued*

mean ± SD. (**B**) Platelet aggregation triggered by hDectin-1 or hPdpn. Human platelets were washed and stimulated with $1\times10^5$ 2B4 cells expressing hDectin-1 (black line) or hPdpn (gray line). (**C**) Surface expression levels of hDectin-1 (hD1) on splenocytes isolated from wild-type (WT) and hD1 Tg mice were analyzed using an isotype control (mouse IgG1 κ, open histograms) or an anti-hDectin-1 antibody BC2 (gray histograms). (**D**) Immunoprecipitated splenocytes from WT and hD1 Tg mice were treated with OpeRATOR. Intact and cleaved hD1A was detected using an anti-hDectin-1 monoclonal antibody (arrowheads). '−': cells without OpeRATOR. These results are representative of three independent experiments. The raw image of western blot was shown in *Figure 6—source data 1*. (**E**) The whole images of E17.5 WT, *Pdpn⁻/⁻*, and hDectin-1 Tg×*Pdpn⁻/⁻* embryos. (**F, G**) Whole-mount triple fluorescence confocal microscopy of embryonic back skin from E17.5 WT, *Pdpn⁻/⁻*, and hDectin-1 Tg×*Pdpn⁻/⁻* embryos using antibodies against LYVE-1 (green), Ter-119 (blue), and PECAM-1 (red). N (**F, G**): WT=4, *Pdpn⁻/⁻*=2, hDectin-1 Tg×*Pdpn⁻/⁻*=4. Representative low and high magnification images are shown in (**F**) and (**G**), respectively. Area enclosed with white dotted-lines in the Ter-119 panels in (**G**) indicates the region of lymphatic vessels. Scale bars, 200 μm in (**F**) and 50 μm in (**G**).

The online version of this article includes the following source data and figure supplement(s) for figure 6:

**Source data 1.** Unedited and labeled Western blot in *Figure 6B*.

**Figure supplement 1.** Role of hDectin-1−CLEC-2 interaction in immune homeostasis.

*and G*). These results suggest that ectopic expression of hDectin-1 in mice rescued the defect of blood-lymph separation caused by Pdpn deficiency.

In the present study, we provide the first example of a PRR regulating organogenesis by serving as a ligand for another receptor.

## Discussion

PRRs were initially identified as host receptors for pathogens but were subsequently found to recognize self-derived components as well. The current study further expands the capacities of PRRs with the demonstration that they can serve as self-ligands for other host receptors; this finding extends the concept of multitasking on the limited number of germline-encoded receptors.

Three examples of heterophilic interactions between CLRs have been reported thus far: NKRP1A−LLT1, NKp80−AICL, and NKp65−KACL (*Rosen et al., 2005*; *Spreu et al., 2010*; *Welte et al., 2006*). Of note, all these CLRs are members of CLR group V which includes Dectin-1 cluster CLRs such as Dectin-1 and CLEC-2 (*Zelensky and Gready, 2005*). Thus, self-recognition may be the original characteristic of CLRs of this group and Dectin-1 might be exceptional in that it has acquired PRR function for pathogens. To fully understand the network shaped by these hitherto unappreciated self-interactions, it will be worthwhile to further investigate potential mutual interactions between other PRRs, particularly CLRs.

**Table 1.** Analysis of offspring derived from the intercross of human Dectin-1 Tg × *Pdpn⁺/⁻* or *Pdpn⁺/⁻*.

Human Dectin-1 Tg × *Pdpn⁺/⁻*, or *Pdpn⁺/⁻* mice were intercrossed. The number of offspring with the *Pdpn⁺/⁺*, *Pdpn⁺/⁻* or *Pdpn⁻/⁻* genotype with the indicated background genotype are shown. Numbers in parentheses indicate the expected number of pups. The significance of differences compared with *Pdpn⁻/⁻* on a wild-type background was evaluated using the chi-square test. *, $0.01 < p < 0.05$. Statistical analysis was performed using JMP 14 software (Statistical Analysis System, Cary, NC).

| | Pdpn | | |
|---|---|---|---|
| Genotype | +/+ | +/− | −/− |
| Wild-type | 36 (24) | 60 (48) | 0 (24) |
| hDectin-1 Tg | 26 (20) | 44 (40) | 10 (20)* |

To date, the stalk (also called neck) region of CLRs has not been regarded as having an active function. Rather, it is presumed to increase the valency of CLRs by self-multimerizing via the coiled-coil, collagen domain, disulfide bond, or heptad repeat (*Beavil et al., 1992*; *Miyake et al., 2015*; *Takahara et al., 2002*; *Zhang et al., 2001*), which may compensate for the known low affinity of CRD ligands. The present study suggests that the stalk region is not merely a multimerizing tool, but also an *O*-glycosylated functional domain. It will be worthwhile to examine whether many other mucin-like proteins possess previously unappreciated *O*-glycosylated motifs that act as specific ligands.

One of the unique features of this CLEC-2 ligand is that its activity could be 'canceled', either by splicing out the stalk-encoding exon or deglycosylation. Indeed, terminal sialic acids of mucin-type glycans can be actively removed by multiple sialidases (*Varki and Gagneux, 2012*).

Although the regulatory process is not fully understood, tissue- and stimuli-specific expression of the hDectin-1A/B isoforms have been reported (*Fischer et al., 2017*; *Willment et al., 2001*; *Willment et al., 2005*) and glycosylation profiles may differ based on cellular status (*Pinho and Reis, 2015*). In this regard, a reported interaction of galactose-specific lectin and Dectin-1 might be mediated based on this principle (*Daley et al., 2017*; *Deerhake et al., 2021*; *Hayen et al., 2018*). Such post-transcriptional modification of ligands may provide an additional layer of regulation of CLR signaling depending on the environmental status.

Dectin-1 cluster genes are widely expressed even in non-immune cells, such as endothelial cells, in sharp contrast to CLRs in the Dectin-2 cluster which are exclusively expressed in immune-related cells (https://www.proteinatlas.org). The inverted half ITAM, called hemITAM, in Dectin-1 appears to have been acquired independently of the canonical ITAM immune module during evolution. Thus, this cluster seems not to be originally acquired as professional immune receptors. Syk-independent functions of Dectin-1 support this idea; one plausible role is phagocytosis, as internalization via Dectin-1 does not require Syk (*Herre et al., 2004*). Functioning as a 'ligand' is additional evidence for a Dectin-1 function independent of Syk in non-immune tissues.

The molecular basis for the pleiotropic function of CLEC-2 in platelets are not fully understood. Although the indispensable role of CLEC-2 in lymphangiogenesis is well-established, currently proposed mechanisms, such as local coagulation in mice (*Bertozzi et al., 2010*; *Hess et al., 2014*; *Uhrin et al., 2010*), cannot fully explain this complicated process (*Herzog et al., 2013*; *Osada et al., 2012*; *Suzuki-Inoue et al., 2017*). In this regard, the identification of a new CLEC-2 ligand that contributes to lymphangiogenesis without inducing coagulation might provide a valuable tool to address the detailed mechanism (*Figure 6—figure supplement 1C*). The apparent weak binding of hDectin-1 to CLEC-2 compared with that of Pdpn may be due to the shorter binding motif of hDectin-1 (EDxxT) compared with that of Pdpn (EDxxxT) or proximal *O*-glycan(s). Alternatively, but not exclusively, different ligands might create different qualities of receptor signaling, by acting as partial agonists/antagonists, which lead to distinct responses, similar to other germline-encoded ligands-receptor systems (*Radtke et al., 2010*).

In contrast to Pdpn, Dectin-1 is also expressed in myeloid lineages in peripheral blood. The weak binding capacity of hDectin-1 may be advantageous in limiting unnecessary 'autonomous' activation of megakaryocytes or platelets through bystander interactions with myeloid cells in the periphery, as no spontaneous thrombosis is observed in the peripheral blood of human and hDectin-1 Tg mice. Furthermore, CLEC-2 did not induce activation of myeloid cells through hDectin-1 (*Figure 6—figure supplement 1D*). Thus, hDectin-1 may be a moderate 'homeostatic' ligand for CLEC-2 that controls physiological processes through platelets beyond thrombosis. Broader distribution of hDectin-1 might provide additional advantages *in vivo*, which could be addressed by tissue-specific Dectin-1-humanized (Tg) mice, ideally on a hCLEC-2- and hPdpn-knock-in background.

To assess this, the analysis on the genetical mutation of hDectin-1 and diseases would be a straightforward approach. However, reported individuals had truncated mutant Dectin-1 (Y238X), which lacks CRD but retains EDxxT motif (*Ferwerda et al., 2009*), implying the importance of CLEC-2 ligand to be detected as a viable variant.

In contrast to well-organized genetical processes, glycosylation is a complex event with many exceptions and glycan structures are diverse depending on species, organs, and developmental stages. The principle organizing this complicated process remains an open question. The future development of spatiotemporal glycomics together with transcriptomics/proteomics at a single-cell level will clarify how germline-encoded receptors have acquired multiple functions.

## Materials and methods

### Materials

Primers, guide RNAs, antibodies, and reagents used in this study are also listed in *Supplementary files 1 and 2*.

### Mice

All animal protocols were approved by the committee of Ethics on Animal Experiment of the Research Institute for Microbial Diseases, Osaka University (Permit number: Biken-AP-R03-17-0). Male and

female mice (7–9 weeks of age) with a C57BL/6 background were used. C57BL/6 mice were purchased from CLEA Japan, Inc (Tokyo, Japan). All mice were maintained in a filtered-air laminar-flow enclosure and given standard laboratory food and water *ad libitum*. CLEC-2-deficient mice and Pdpn-deficient mice were established using the CRISPR-Cas9 system to modify the genomes of C57BL/6 embryos. The CRISPR Design Tool (**Naito et al., 2015**) was used to design gRNAs (**Figure 2—figure supplement 1A** and **Figure 6—figure supplement 1A**), which were cloned into the BbsI site of plasmid pX330 (Addgene). Potential CRISPR-edited alleles were PCR-amplified and directly sequenced. Alternatively, amplicons were electrophoresed using TBE-based 4% polyacrylamide gels to discriminate wild-type from mutant alleles (**Zhu et al., 2014**). To reduce the off-targeting effects, we used two different Pdpn-deficient lines. For establishment of the *Clec1b*$^{\Delta HPC}$ mice, livers from *Clec1b*$^{-/-}$ fetuses (E15.5), which contain hematopoietic progenitor cells (HPCs), were collected, and then dissociated into single-cell suspensions. Cells were treated with ammonium chloride solution to remove erythrocytes and mesh filtered. Next, $1\times10^6$ cells were intravenously administered to CD45.1$^+$ recipient mice irradiated with 8 Gy. After 8 weeks, peripheral blood cells were stained with anti-CD45.1 and anti-CD45.2 antibodies. Reconstitution was determined according to the ratio of CD45.2$^+$ to CD45.1$^+$ cells. For establishment of the hDectin-1 transgenic (Tg) mice, hDectin-1 cDNA was inserted into the EcoRI site of pCAGGS vector (**Niwa et al., 1991**), and hDectin-1 Tg mouse lines were produced by injecting the linearized vector into fertilized eggs of C57BL/6 mice. PCR-based genotyping (primers listed in **Supplementary file 1**) was performed using genomic DNA isolated from mouse tails as a template. To produce hDectin-1 Tg×*Pdpn*$^{-/-}$ mice, hDectin-1 Tg×*Pdpn*$^{+/-}$ mice were intercrossed.

## Cell lines

All cell lines used were generated in our lab. Plasmids used for transfections were verified by Sanger sequencing prior to transfections and successful infections were confirmed via flow cytometry with antibodies against the target proteins. Knockout cell lines were generated via CRISPR and successful knockout was confirmed by direct sequencing. All cell lines were tested for mycoplasma contamination before generation.

The 2B4 NFAT-GFP reporter cells expressing the CLRs CLEC-2 and Dectin-1 were prepared as previously described (**Miyake et al., 2013**; **Toyonaga et al., 2016**; **Yamasaki et al., 2008**). HeLa cells expressing hDectin-1, C1GalT1-specific chaperone (COSMC), and Core 2 β1,6 *N*-acetylglucosaminyltransferase (C2GnT), NeuAcα2-3Galβ1-3GalNAcα2–6 sialyltransferase (ST6GalNAc4) were established by lipofection using pMX retroviral vectors. Genes encoding human COSMC and ST6GalNAc4 were separately incorporated into pMX-IRES-rCD2 and pMX-IRES-hCD8 vectors, respectively. The pMX-COSMC-IRES-rCD2 vector was transfected into COSMC-deficient HeLa cells by lipofection. The pMXs-C2GnT-IRES-hCD8 and pMX-ST6GalNAc4-IRES-hCD8 vectors were separately transfected into HeLa cells expressing hDectin-1 by lipofection. The pMXs-C2GnT-IRES-hCD8 vector was kindly provided by Dr. Hisashi Arase (Osaka University, Japan). COSMC-deficient cells were established as follows: Target sites on human COSMC were determined using a CRISPR Design Tool (**Naito et al., 2015**) that minimizes exonic off-target effects. gRNAs (**Supplementary file 1**) were cloned into the lentiCRISPRv2 puro vector (Addgene) digested with BsmBI. A COSMC-deficient cell line was generated through transient transfection of HeLa cells expressing hDectin-1 with lentiCRISPRv2 puro vectors containing target sequences. After 4 days, single-cell clones were isolated from mass cultures using the limiting dilution technique.

## Reagents

Cell trace violet (C34571) was purchased from Thermo Fisher Scientific. PHZ hydrochloride (114715), zymosan (Z4250), Acid citrate-dextrose solution (C3821), and HAT Media Supplement (H0262-10VL) were purchased from Sigma-Aldrich. PNGase F (P0704S), Deglycosylation Mix (P6039S), BbsI (R0539), and BsmBI (R0580) were purchased from New England Biolabs. EcoRI (1040B) was purchased from TAKARA Bio. Biotinylated Peanut agglutinin (PNA) (BA-0074), biotinylated Wheat germ agglutinin (WGA) (B-1025), and biotinylated Maackia amurensis lectin II (MAL-II) (B-1315) were purchased from VECTOR Laboratories. OpeRATOR (G2-OP1-020) and SialEXO (G2-OP1-020) were purchased from Genovis. Neuraminidase from Arthrobacter ureafaciens (24229-74), Lymphocyte Separation Solution (d=1.077) (20828-15), and Chemi-Lumi One Super (02230-30) were purchased from Nacalai Tesque. Prostaglandin I2 (61849-14-7) was purchased from Cayman Chemical. TiterMax Gold (G-1)

was purchased from TiterMax. PEG1500 (10783641001) and 1% BM-Condimed (11088947001) were purchased from Roche. QIAzol Lysis Reagent (79306) was purchased from QIAGEN. Mouse anti-human CD14 MicroBeads (130-050-201) was purchased from Miltenyi Biotech. Tissue-Tek OCT compound (4583) was purchased from Sakura Finetek.

### *In vitro* cell-based assays

The 2B4 NFAT-GFP reporter cells expressing CLEC-2 or Dectin-1 were prepared as previously described (*Miyake et al., 2013*; *Toyonaga et al., 2016*; *Yamasaki et al., 2008*). Details of each experiment using reporter cells and/or platelets are summarized in the following sections.

### Preparation of human and mouse peripheral blood cells

Collection and use of human peripheral blood cells were approved by the Institutional Review Board of the Research Institute for Microbial Diseases, Osaka University (Permit number: 29-12). Human venous blood was collected from healthy volunteers who given informed consent and consent to publish. Consent documents and procedures were approved by the Institutional Review Board of the Research Institute for Microbial Diseases, Osaka University (Permit number 29-12). Venous blood from healthy drug-free volunteers was collected into 10% sodium citrate. Platelet-rich plasma (PRP) was prepared using centrifugation at 400×*g* for 10 min at room temperature. Human monocytes were prepared as previously described (*Kiyotake et al., 2015*). Briefly, peripheral blood mononuclear cells (PBMCs) from a healthy donor were isolated using gradient centrifugation with Lymphocyte Separation Solution (d=1.077) (Nacalai Tesque, Kyoto, Japan). CD14$^+$ monocytes were purified from PBMCs using anti-human CD14 MicroBeads (Miltenyi Biotech, Bergisch Gladbach, Germany). Fractionated CD14$^-$ or CD14$^+$ cells were stained with an anti-human CD14 antibody and anti-hDectin-1A antibody designated BC2. The Institutional Review Board of the Research Institute for Microbial Diseases, Osaka University approved the collection and use of human PBMCs.

Wild-type and *Clec1b*$^{\Delta HPC}$ mice were killed using $CO_2$, and 900 µl of blood was drawn using heart puncture and collected into 100 µl of acid citrate-dextrose. For density gradient separation of peripheral blood cells, whole blood was treated with ammonium-chloride-potassium (ACK), and then separated using centrifugation at 400 rpm for 10 min. To separate platelets, PRP was prepared using centrifugation at 100×*g* for 10 min at room temperature. PRP was further centrifuged at 400×*g* for 10 min in the presence of prostaglandin $I_2$ (1 µg/ml) (Cayman Chemical, Ann Arbor, MI) to obtain washed platelets, which were resuspended in modified Tyrode's buffer (*Suzuki-Inoue et al., 2003*) at the indicated cell densities.

### *In vitro* stimulation assay using reporter cells

CellTrace Violet (Thermo Fisher Scientific, Waltham, MA)-labeled 2B4 NFAT-GFP reporter cells (3×10$^4$) were treated with tissue homogenates from wild-type or Rag1-deficient mice, blood cells derived from humans or mice, CLR-expressing cells, or 10 µg/ml zymosan (Sigma-Aldrich, St. Louis, MO). After 18-h incubation, the fluorescence of NFAT-GFP was monitored using a flow cytometer.

### Mouse model of PHZ-induced hemolytic anemia

Treatment of PHZ to induce hemolytic anemia was performed as described previously (*Tolosano et al., 2002*). Briefly, PHZ (150 µg/g body weight) was dissolved in 200 µl sterile phosphate-buffered saline (PBS). Wild-type mice were injected intraperitoneally with a single dose of PHZ-PBS solution or PBS alone. The spleens were collected from anesthetized mice 3 days after the treatment.

### Generation of an anti-human platelet monoclonal antibody (6D11)

To generate an anti-human platelet antibody, a Wistar rat (CLEA Japan) was immunized intraperitoneally with human platelets two times. TiterMax Gold (TiterMax, GA) was also injected intraperitoneally at the initial immunization. For the final immunization, the Wistar rat was immunized subcutaneously in the foot pad with human platelets emulsified in the adjuvant. Lymph node cells were fused with NSO$^{bcl2}$ myeloma cells (*Ray and Diamond, 1994*) using polyethylene glycol 1500 (Roche Diagnostics, Mannheim, Germany). Hybridoma cells were selected in Dulbecco's Modified Eagle Medium (DMEM) containing 10% fetal bovine serum (FBS), hypoxanthine-aminopterin-thymidine (Sigma-Aldrich), and 1% BM-Condimed (Roche Diagnostics).

The supernatants of hybridomas that specifically inhibited the interaction between hDectin-1 and human platelets were selected and then evaluated using immunoprecipitation with human platelets. We isolated the best monoclonal antibody designated 6D11. The nucleotide sequences of the complementarity determining regions (CDRs) of the immunoglobulin gene of this hybridoma were determined using the technique of 5'-rapid amplification of cDNA ends (5'-RACE) (*Frohman et al., 1988*) at Frontier Institute (Hokkaido, Japan). Heavy and light chain sequences were inserted into the pMX-IRES-hCD8 and pMX-IRES-hCD4 vectors, respectively. Recombinant 6D11 was expressed by HEK293 cells and purified using a spin column-based Antibody Purification Kit (COSMO BIO, Tokyo, Japan).

## Identification of 6D11-interacting proteins

Experimental details of generation of an anti-human platelet monoclonal antibody (6D11) are provided in *Supplemental Methods*. Washed human platelets ($1.75\times10^{10}$) were lysed in Nonidet P-40 (NP-40, Nacalai Tesque, Kyoto, Japan) lysis buffer. Lysates were incubated with protein-G sepharose beads (Cytiva, Marlborough, MA) at 4°C for 1 hr, followed by centrifugation (repeated five times). The supernatant was mixed with beads and subsequently incubated with an isotype control (rat IgG1 κ, BD Pharmingen, San Diego, CA) or biotinylated 6D11 (4°C, 1 hr). The beads were collected using centrifugation and washed five times with NP-40 lysis buffer and two times with 0.1 M $NH_4HCO_3$, followed by elution with 40 µl of 0.1 M glycine-HCl (pH 2.0). After neutralization, bound proteins were eluted with 5.7 µl of SDS sample buffer by heat denaturation. Details regarding the mass spectroscopic analysis are provided in the next section.

## Mass spectroscopic analysis of identification of 6D11-interacting proteins

Methanol and chloroform were added to the eluted buffer for protein precipitation followed by dissolution into Rapigest solution (Waters Corporation, Milford, MA). The protein solution was subjected to reduction with dithiothreitol, followed by alkylation with iodoacetamide, digestion by trypsin and purification with C18 tip (GL-SCIENCE, Tokyo, Japan). The trypsinized and purified solution was subjected to LC-MS/MS, Bruker TEN column (Bruker, Bremen, Germany) on a Nano Elute nanoLC system coupled with timsTOF Pro mass spectrometer (Bruker). The column temperature was set to 50°C. The mobile phase consisted of water containing 0.1% formic acid (solvent A) and acetonitrile containing 0.1% formic acid (solvent B). The peptides were eluted with the gradient setting of 2–35% solvent B for 18 min at the flow rate of 500 nl/min. The mass scanning range was set from 300 to 1700 in *m/z* and the ion mobility revolution mode was set to custom range from 0.85 to 1.30 in Vs/cm². The ion spray voltage was set at 1.6 kV in the positive ion mode. The MS/MS spectra were acquired by automatic switching between the MS and MS/MS modes. Data Analysis (Bruker) software was used for processing the mass data. Peptides were identified by database searching using the MASCOT Server (Matrix Science, London, UK). Precursor mass tolerance was set to 15 ppm and fragment mass tolerance was set to 0.05 Da. Carbamidomethylation of cysteine was set as static modification. Oxidation of methionine, acetyl of protein N-terminus, and deamination of asparagine and glutamine were set as variable modification. The database for searching was SwissProt (2019_08) or taxonomy-limited by *Homo sapiens*. The search results were analyzed by Scaffold4 (Proteome Software, Portland, USA).

## CLEC-2 internalization assay

Washed murine platelets ($1\times10^5$) were incubated with 2B4 cells expressing CLRs for 18 hr, and the surface expression of CLEC-2 was detected using a PE-conjugated anti-mouse CLEC-2 antibody. Fluorescence of PE was monitored using a flow cytometer.

## Immunoblotting analysis

2B4 cells expressing hDectin-1A or 1B ($2\times10^6$ for glycosidase analysis, $1\times10^8$ for OpeRATOR treatment) and splenocytes ($6\times10^7$) isolated from hDectin-1 transgenic mice were lysed with NP-40 lysis buffer. The solubilized hDectin-1 was captured on beads with an anti-hDectin-1 mAb (MAB1859, R&D Systems, Minneapolis, MN). For glycosidase analysis, the immobilized proteins were treated with PNGase F or PNGase F plus *O*-glycosidase mix including Neuramindase, Hexosaminidase, and *O*-glycosidase (New England Biolabs, Beverly, MA) according to the manufacturer's instruction. *O*-glycosylated peptides

were liberated by OpeRATOR and SialEXO (Genovis, Lund, Sweden) according to the manufacturer's instructions, electrophoresed through 10% acrylamide (Nacalai Tesque) or 10–20% gradient gels (ATTO, Tokyo, Japan) and blotted onto polyvinylidene difluoride (PVDF) membranes (Millipore, Billerica, MA). PVDF membranes were detected using anti-hDectin-1 polyclonal antibodies (AF1859, R&D Systems), followed by incubation with an HRP-labeled secondary IgG (Invitrogen, Carlsbad, CA) or Protein G (Sigma-Aldrich, St. Louis, MO). Immunocomplexes were detected using Chemi-Lumi One Super (Nacalai Tesque) and a Luminescent Image Analyzer LAS-1000 (Fujifilm, Tokyo, Japan).

## Sialidase treatment

2B4 cells expressing hDectin-1 were incubated with 2B4 cells that expressed hCLEC-2 or 10 µg/ml zymosan in the absence or presence of *Arthrobacter ureafaciens* sialidase (50 mU/ml) (Nacalai Tesque) at 37°C for 18 hr. After incubation, the cytoplasmic fluorescence of GFP and that of surface glycans was detected using 5 µg/ml biotinylated PNA and MAL-II lectins (VECTOR Laboratories, Burlingame, CA) were measured using a FACS Calibur flow cytometer (BD Biosciences, San Jose, CA).

## Structural analysis of *O*-linked glycans on human Dectin-1

2B4 cells expressing hDectin-1 were lysed with NP-40 lysis buffer. The lysate was mixed with beads and an anti-hDectin-1 mAb (MAB1859), subjected to SDS-PAGE and stained with Coomassie Brilliant Blue. hDectin-1 corresponding bands were excised from the gel and the gel strips were incubated with trypsin and endoprotease Glu-C (37°C, 16 hr) after reduction with dithiothreitol and alkylation with iodoacetamide. Peptides and glycopeptides extracted from the digests were analyzed using LC-ESI MS.

## Mass spectroscopic analysis of *O*-linked glycans on human Dectin-1

The peptides and glycopeptides were separated using a Develosil 300 ODS-HG-5 column (150×1.0 mm$^2$ i.d., Nomura Chemical). The mobile phases were as follows: (A) 0.08% formic acid and (B) 0.15% formic acid/80% acetonitrile. The column was eluted with solvent A for 5 min, after which the concentration of solvent B was increased to 40% for 55 min (50 µl/min) using an Accela HPLC system (Thermo Fisher Scientific). The eluate was continuously introduced into an ESI source, and the peptides and glycopeptides were analyzed using an LTQ Orbitrap XL (Hybrid Linear Ion Trap-Orbitrap Mass Spectrometer; Thermo Fisher Scientific). MS settings were as follows: voltage of the capillary source was set to 4.5 kV, and the temperature of the transfer capillary was maintained at 300°C. The capillary and tube lens voltages were set to 15 and 50 V, respectively. MS data were obtained in positive-ion mode (*m/z* 300–3000 resolution, 60,000; mass accuracy, 3 ppm). MS/MS data were obtained using an Ion Trap LTQ Orbitrap XL (data-dependent top 3, CID). To detect and quantitate the levels of the hDectin-1 glycoforms of each *O*-glycosylation site, samples extracted from the ion chromatogram (EIC) were analyzed according to their theoretical monoisotopic masses (GlycoMod tool, http://web.expasy.org/glycomod/, mass tolerance; ±0.003 Da) of doubly charged ions [M+2H]$^{2+}$ (z=2) of the corresponding glycopeptide using Xcalibur software ver. 2.2 (Thermo Fisher Scientific).

## Surface plasmon resonance

SPR assays were performed using a BIACORE T-200 (Cytiva, MA, formerly GE Healthcare). The *O*-glycosylated hDectin-1 peptide harboring S98-K108 (disialyl-core 1 *O*-glycan was attached onto T105, NH$_2$-SSLEDSV[NeuAcα2-3Galβ1-3(NeuAcα2–6)GalNAcα1-*O*-T$^{105}$]PTK-COOH) was chemoenzymatically synthesized as described previously (*Yoshimura et al., 2019*). The non-glycosylated hDectin-1 peptide encoding the same region was purchased from Genscript. Recombinant CLEC-2 carbohydrate-recognition domain (CRD) was immobilized through amine coupling to a CM5 sensor chip (Cytiva). A series of (glyco)peptides (0, 3.125, 6.25, 12.5, 25, 50, 100, and 200 µM) were injected at the flow rate of 30 µl/min.

## Modeling of CLEC-2 CRD–hDectin-1 complex

The structure of hDectin-1 was modeled using the homology modeling approach. We used structures of murine Dectin-1 (PDB ID: 2BPE) (*Brown et al., 2007*), activating Ly49H receptor (PDB ID: 4JO8) (*Berry et al., 2013*), and the fragment of DC-SIGNR (PDB ID: 1SL6) (*Guo et al., 2004*) as the template structure. A total of 200 structures were modeled by the Modeller program (version 9.25).

Three disulfide bonds were assigned during the homology modeling, and thorough VTFM optimization (repeating the whole cycle five times) was performed. All 200 models were assessed by the DOPE score, and the one with the lowest dope score was used for modeling interaction with C-type lectin-like receptor CLEC-2 (PDB ID: 3WSR) (*Nagae et al., 2014*). The structure of *O*-glycan (Galβ1-3[NeuAcα2–6]GalNAcα) was generated using GLYCAM-web (http://glycam.org/) and attached to Thr manually. Following this, glycosylated hDectin-1 was manually docked to CLEC-2 by superimposing the *O*-glycan of hDectin-1 over the *O*-glycan of Pdpn in CLEC-2−Pdpn crystal structure (PDB ID: 3WSR). Coordinated glycosylated hDectin-1 in docked conformation were extracted and merged into the crystal structure coordinates of CLEC-2. The overall complex was subjected to energy minimization and extensive molecular dynamics simulation to allow sampling of the binding mode between CLEC-2 and hDectin-1.

The input structure for molecular dynamics simulation was prepared using the *tleap* module of AmberTools20. The proteins and glycans were described by AMBER ff14SB and GLYCAM (version 06j) force fields, respectively. All initial structures were solvated in an octahedral box of TIP3P water molecules extending 12 Å in each direction using *tleap*. Two sodium ions were added to neutralize the charge of this system. The CLEC-2−Dectin-1 complex was equilibrated using the multistep equilibration protocol published elsewhere (*Nagae et al., 2017*). A 100-ns MD simulation at NPT was then performed for each case using CUDA implementation of the *pmemd* from AMBER20. To confirm the binding of *O*-glycan in CLEC-2, MD simulations were extended to 100 ns. During the MD, the temperature was kept constant at 300K using the Berendsen weak-coupling method. The time constant for heat bath coupling was set to 5 ps. The Particle Mesh Ewald (*Darden et al., 1993*) was used for calculating the electrostatic forces. A cutoff of 9 Å was used for nonbonded interactions, and the SHAKE algorithm (*Ryckaert et al., 1977*) was used to restrain the hydrogen atoms. The MD trajectories were analyzed (hydrogen bond analysis and calculation of dihedral angles) using *cpptraj* utility of AmberTools20 (*Roe and Cheatham, 2013*). The final MD pose was interpreted and presented in a figure.

## Platelet aggregation assay

Human PRP was stimulated with 2B4 cells expressing hDectin-1 or human Pdpn ($1 \times 10^5$ cells). Platelet aggregation was monitored by measuring light transmission with the use of PRP313M aggregometer (TAIYO Instruments INC) for 15 min under constant stirring at 1000 rpm at 37°C.

## Immunohistochemical analysis

Whole E17.5 embryos were dissected and fixed overnight at 4°C in 4% paraformaldehyde (PFA)/PBS (Nacalai Tesque). The skin on the back was peeled off and fixed overnight in 4% PFA/PBS at 4°C. Tissues were washed in PBS with 0.2% Triton X-100 (PBS/T) at 4°C for 30 min two times, blocked in PBS/T containing 1% (w/v) goat serum albumin at room temperature for 1 hr, and incubated with primary antibodies (American hamster anti-mouse PECAM-1, rabbit anti-mouse LYVE-1, and rat anti-mouse TER-119) in blocking solution at 4°C overnight. These tissues were further washed three times in PBS/T for 30 min at 4°C and two times at room temperature, followed by overnight incubation with secondary antibodies (Cy3-conjugated anti-Armenian hamster IgG, Cy5-conjugated anti-rat IgG, and Alexa Fluor 488-conjugated anti-rabbit IgG) in blocking solution at 4°C. Tissues were washed three times in PBS/T for 30 min at 4°C and two times at room temperature.

For the observation of the junction of cardinal veins and lymphatic sacs, whole E13.5 embryos were dissected and fixed for 24 hr at 4°C in 4% PFA/PBS, then kept 24 hr in 30% sucrose for dehydration. Embryos embedded in Tissue-Tek OCT compound (Sakura Finetek, Tokyo, Japan) were snap-frozen in liquid nitrogen and 10 μm cryostat sections were prepared. Slices were treated with 0.1% trypsin at 37°C for 15 min and blocked in PBS containing 5% goat serum and 0.3% Triton X-100 at room temperature for 1 hr. For immunofluorescence, slices were incubated with primary antibodies (American hamster anti-mouse PECAM-1, Rabbit anti-mouse LYVE-1, and Rat monoclonal anti-mouse CD41) in blocking solution at 4°C overnight. These slices were washed three times in PBS for 3 min at room temperature, followed by 1 hr incubation with secondary antibodies (Alexa Fluor 488-conjugated anti-Armenian hamster IgG, Alexa Fluor 555-conjugated anti-rat IgG, and Alexa Fluor 647-conjugated anti-rabbit IgG) in PBS containing 5% goat serum and 1% BSA at room temperature. Tissues were washed three times in PBS for 3 min at room temperature.

## Microscopy

Embryos (E17.5) were photographed at autopsy, and the skin on the back was subjected to immunohistochemical analysis as described previously (*Fu et al., 2008*; *Uhrin et al., 2010*). For the observation of the junction of cardinal veins and lymphatic sacs, whole E13.5 embryos were dissected and subjected to immunohistochemical analysis as summarized in *Supplemental Methods*. Confocal microscopy was performed using an Olympus FV-10i (Olympus, Tokyo, Japan). Separate green, red, and blue images were collected and analyzed with Fiji ImageJ software (*Schindelin et al., 2012*). Maximum intensity projections of z-stack images were presented in each figure.

## RNA sequencing

Human monocytes ($5\times10^5$ cells) were left untreated or stimulated with murine platelets from wild-type or *Clec1b*$^{\Delta HPC}$ ($1\times10^7$ cells) mice for 8 hr at 37°C. Total RNA was extracted from the cells with a QIAzol Lysis Reagent (Qiagen, Hilden, Germany) according to the manufacturer's instructions. Library preparation was performed using a TruSeq stranded mRNA sample prep kit (Illumina, San Diego, CA) according to the manufacturer's instructions. Whole transcriptome sequencing was applied to the RNA samples by using Illumina HiSeq 2500, HiSeq 3000, or Novaseq 6000 platforms in a 75- or 101-base single-end mode. Sequenced reads were mapped to the human reference genome sequences (hg19) using TopHat ver. 2.0.13. The number of fragments per kilobase of exon per million mapped fragments (FPKMs) was calculated using Cufflinks ver. 2.2.1. Unsupervised k-means clustering analysis was performed by integrated Differential Expression and Pathway analysis (iDEP) website. The transcriptome RNA-sequencing datasets have been deposited to the National Center for Biotechnology Information Gene Expression Omnibus database under the accession number GSE196049.

## Statistical analysis

An unpaired two-tailed Student t-test was used for all statistical analyses.

## Acknowledgements

The authors are grateful to D Motooka, S Iwai, F Sugihara, K Kaseda, M Ikawa, H Shimizu, K Matoba, W Okawa, and D Murai for technical support; M Netea, Y Kizuka, Y Yamaguchi, N Taniguchi, J Takagi, and Y Ogawa for discussion. The authors also thank the Cooperative Research Project Program of the Medical Institute of Bioregulation, Kyushu University, the NGS core facility of Genome Information Research Center, Research Institute for Microbial Diseases, Osaka University for their technical support. This work was supported by JSPS KAKENHI Grant numbers JP 16H06276 (AdAMS), 20H00505, 20K06575, and 22H05183 and by the Grant for Joint Research Project of the Research Institute for Microbial Diseases, and IFReC Kishimoto Foundation Fellowship, Osaka University.

## Additional information

### Funding

| Funder | Grant reference number | Author |
| --- | --- | --- |
| Japan Society for the Promotion of Science | 20H00505 | Sho Yamasaki |
| Japan Society for the Promotion of Science | 20K06575 | Masamichi Nagae |
| Japan Society for the Promotion of Science | 16H06276 | Sho Yamasaki |
| Japan Society for the Promotion of Science | 22H05183 | Sho Yamasaki |

The funders had no role in study design, data collection and interpretation, or the decision to submit the work for publication.

## Author contributions
Shojiro Haji, Conceptualization, Investigation, Writing – original draft; Taiki Ito, Takashi Shimizu, Daiki Mori, Sushil K Mishra, Investigation; Carla Guenther, Masamichi Nagae, Investigation, Writing – original draft, Writing - review and editing; Miyako Nakano, Investigation, Writing – original draft; Yasunori Chiba, Janet A Willment, Gordon D Brown, Resources; Masato Tanaka, Resources, Investigation; Sho Yamasaki, Conceptualization, Supervision, Funding acquisition, Writing – original draft, Project administration, Writing - review and editing

## Author ORCIDs
Shojiro Haji http://orcid.org/0000-0003-3651-2836
Taiki Ito http://orcid.org/0000-0002-1408-7288
Carla Guenther http://orcid.org/0000-0001-7915-1028
Sushil K Mishra http://orcid.org/0000-0002-3080-9754
Janet A Willment http://orcid.org/0000-0002-7040-0857
Masamichi Nagae http://orcid.org/0000-0002-5470-3807
Sho Yamasaki http://orcid.org/0000-0002-5184-6917

## Ethics
Human subjects: All human subjects research was approved by the Institutional Review Board of the Research Institute for Microbial Diseases, Osaka University. Informed consent and consent to publish were obtained from all individuals donating venous blood. Consent documents and procedures were approved by the Institutional Review Board of the Research Institute for Microbial Diseases, Osaka University (Permit number 29-12).

All animal protocols were approved by the committee of Ethics on Animal Experiment and Research Institute for Microbial Diseases, Osaka University (Permit number: Biken-AP-R03-17-0).

## Decision letter and Author response
Decision letter https://doi.org/10.7554/eLife.83037.sa1
Author response https://doi.org/10.7554/eLife.83037.sa2

# Additional files

## Supplementary files
- Supplementary file 1. List of primers and gRNAs used in this study.
- Supplementary file 2. List of antibodies used in this study.
- MDAR checklist

## Data availability
Sequencing data have been deposited in GEO under accession code GSE196049.

The following dataset was generated:

| Author(s) | Year | Dataset title | Dataset URL | Database and Identifier |
| --- | --- | --- | --- | --- |
| Haji S, Nagae M, Ito T, Motooka D, Okuzaki D, Yamasaki S | 2022 | RNA sequencing of human monocytes stimulated with murine platelets from wild-type or Clec1b-/- mice | https://www.ncbi.nlm.nih.gov/geo/query/acc.cgi?acc=GSE196049 | NCBI Gene Expression Omnibus, GSE196049 |

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
