## [Editor Report]

The C-type lectin receptor family recognises pathogens and self-components. Dectin-1 is known to recognize glucan on pathogens. In this fundamental study Dectin-1 and CLEC2 – another C-type lectin receptor expressed on platelets – interacts through an *O*-glycosylated ligand presented in the stalk region of Dectin-1. This compelling study demonstrates a potential role for pattern recognition receptors in physiological processes.

---

## [Decision Letter]

**Decision letter after peer review:**

Thank you for submitting your article "Human Dectin-1 is *O*-glycosylated and serves as a ligand for C- type lectin receptor CLEC-2" for consideration by *eLife*. Your article has been reviewed by 2 peer reviewers, and the evaluation has been overseen by a Reviewing Editor and Satyajit Rath as the Senior Editor. The following individuals involved in the review of your submission have agreed to reveal their identity: Gavin J Wright (Reviewer #1); Sandra van Vliet (Reviewer #2).

Below we summarise the essential revisions required:

1. The authors use 2B4 transfectants to establish their proposition, but show in figure 2I that human monocytes can interact. Although a full characterization of all human Dectin-1 positive immune subsets would be out of scope for this article please include a basic glycan analysis of Dectin-1 in the human monocytes to confirm that also in these cells Dectin-1 harbors the sialylated Core 1 structure.

2. Figure 2H is not showing CLEC-2 internalization. The authors incubate platelet with plate-bound Dectin-1 transfectants. For CLEC-2 to internalize the Dectin-1 would have to be released from the transfected cells, of which there is no evidence. Therefore, the read-out platelet activation, CLEC-2 shedding or perhaps occupancy of the CLEC-2 receptor. These conclusions should be toned-down or confirmed using time-lapse experiments with pre-labeled CLEC-2 antibodies to convincingly demonstrate that the CLEC-2 is indeed internalized.

3. Which downstream signaling pathway of human Dectin-1 is triggered upon the CLEC-2 interaction? Have the authors performed western blots addressing the phosphorylation of SYK and/or Raf-1?

4. Ferweda et al., described four women with a human Dectin-1 deficiency (NEJM 2009). These women actually suffer from recurrent fungal infections, yet show no other apparent lymphatic defects. How do the authors reconcile their findings with these patients? Can the authors comment on this in their discussion?

5. The authors mainly use an NFAT-GFP reporter cell line as a proxy for hDectin-1 binding. When referring to this activity it would be preferable to state this more precisely rather than say "interacts with hDectin-1" examples on lines 93 and 97.

6. A few of the graphs were difficult to read the x-axis e.g. Figure 1G – requires more space or remove some of the major ticks.

7. The SPR data in Figure 5F is not convincing and does not affect any of the conclusions, but this could be improved. The amount of binding signal for the positive (just 3RU) is very small. I appreciate you are using low molecular mass peptides for this but designing the experiment with a higher mass analyte would give better data. Why not immobilise the peptides and use the CLEC-2 CRD as the analyte? Or a longer peptide?

*Reviewer #1 (Recommendations for the authors):*

My only comments are:

The authors mainly use an NFAT-GFP reporter cell line as a proxy for hDectin-1 binding. When referring to this activity it would be better to state this more precisely rather than say "interacts with hDectin-1" examples on lines 93 and 97.

A few of the graphs were difficult to read the x-axis e.g. Figure 1G – just need more space or take out some of the major ticks.

The SPR data in Figure 5F is not convincing. I don't think this affects any of the conclusions but this could be improved. The amount of binding signal for the positive (just 3RU) is very small. I appreciate you are using low molecular mass peptides for this but designing the experiment with a higher mass analyte would give better data. Why not immobilise the peptides and use the CLEC-2 CRD as the analyte? Or a longer peptide?

*Reviewer #2 (Recommendations for the authors):*

First I would like to compliment the authors on this excellent work. The novelty of the Dectin-1 and CLEC-2 interaction is high and the thorough characterization is done in high molecular detail. In my opinion, the authors convincingly show that the Dectin-1 – CLEC-2 interaction leads to 2-way signaling in a glycan (sialylated Core 1)-dependent manner. Moreover, they demonstrate the functionality of this interaction in a human Dectin1- transgenic mouse model deficient for the CLEC-2 ligand podoplanin.

I have a few suggestions to further strengthen the impact of this manuscript:

1. My only reserve is the relevance of the Dectin-1 – CLEC-2 interaction in humans. The authors mainly use 2B4 transfectants to establish their claims, but do show in figure 2I that human monocytes can interact. Although a full characterization of all human Dectin-1 positive immune subsets would be out of scope for this article, I strongly recommend including a basic glycan analysis of Dectin-1 in the human monocytes to confirm that also in these cells Dectin-1 harbors the sialylated Core 1 structure.

2. I would like to argue that figure 2H is not showing CLEC-2 internalization. The authors incubate platelet with plate-bound Dectin-1 transfectants. For CLEC-2 to internalize the Dectin-1 would have to be released from the transfected cells, of which there is no evidence. In my opinion, the authors read out platelet activation, CLEC-2 shedding, or perhaps occupancy of the CLEC-2 receptor. These conclusions should be toned-down or confirmed using time-lapse experiments with pre-labeled CLEC-2 antibodies to convincingly demonstrate that the CLEC-2 is indeed internalized.

3. Which downstream signaling pathway of human Dectin-1 is triggered upon the CLEC-2 interaction? Have the authors done western blots addressing the phosphorylation of SYK and/or Raf-1?

4. Ferweda et al. described four women with a human Dectin-1 deficiency (NEJM 2009). These women actually suffer from recurrent fungal infections, yet show no other apparent lymphatic defects. How do the authors reconcile their findings with these patients? Can the authors comment on this in their discussion?

---

## [Author Response]

Below we summarise the essential revisions required:1. The authors use 2B4 transfectants to establish their proposition, but show in figure 2I that human monocytes can interact. Although a full characterization of all human Dectin-1 positive immune subsets would be out of scope for this article please include a basic glycan analysis of Dectin-1 in the human monocytes to confirm that also in these cells Dectin-1 harbors the sialylated Core 1 structure.

In accordance with the reviewer’s suggestion, we collected human myeloid cells from peripheral blood and performed a basic glycan analysis. First, human primary myeloid cells were immunoblotted with anti-human Dectin-1. As observed in 2B4 transfectants (original Figure 4A and 4B), the apparent molecular mass of hDectin-1 was higher (approximately 40 kDa) than the expected size of core protein (27 kDa), suggesting that Dectin-1 is glycosylated similarly in human primary cells (see Author response image 1).

**Author response image 1. sa2fig1:** 

In addition, we treated human monocytes with sialidase and co-cultured them with CLEC-2-expressing reporter cells. CLEC-2 reporter cell activity induced by CD14^+^ human monocytes was impaired by this treatment (see Author response image 2), suggesting that human Dectin-1 expressed on human monocytes is sialylated.

2. Figure 2H is not showing CLEC-2 internalization. The authors incubate platelet with plate-bound Dectin-1 transfectants. For CLEC-2 to internalize the Dectin-1 would have to be released from the transfected cells, of which there is no evidence. Therefore, the read-out platelet activation, CLEC-2 shedding or perhaps occupancy of the CLEC-2 receptor. These conclusions should be toned-down or confirmed using time-lapse experiments with pre-labeled CLEC-2 antibodies to convincingly demonstrate that the CLEC-2 is indeed internalized.

We apologize for our misleading interpretation regarding the CLEC-2 internalization. As the reviewer pointed out, the original Figure 2H only showed the downregulated expression of CLEC-2 at 18 h. We therefore corrected our misleading description in the original sentences in the Results section of the revised manuscript (page 6, line 124).

3. Which downstream signaling pathway of human Dectin-1 is triggered upon the CLEC-2 interaction? Have the authors performed western blots addressing the phosphorylation of SYK and/or Raf-1?

We apologize for not clearly describing the direction of the signaling triggered by human Dectin-1 and CLEC-2. As we have shown in the original Figure S6D, CLEC-2 cannot trigger signal through human Dectin-1 in the co-culture system using platelets and primary myeloid cells. Therefore, we did not detect the evidence of human Dectin-1 signaling upon CLEC-2 binding. To avoid the misleading description, we carefully corrected the corresponding sentences in the Discussion sections (page 13, line 317) of the revised manuscript.

4. Ferweda et al., described four women with a human Dectin-1 deficiency (NEJM 2009). These women actually suffer from recurrent fungal infections, yet show no other apparent lymphatic defects. How do the authors reconcile their findings with these patients? Can the authors comment on this in their discussion?

We thank the reviewer for raising this important point. We confirmed, that the position of the missense mutations found in all these patients were in the CRD (Y238X). As human Dectin-1 is a type II membrane protein, this truncated mutant Dectin-1 retained an intact stalk region and can be expressed on the cell surface as reported (Figure S4C, Ferweda, *N Engl J Med*, 2009). From this aspect, this mutant Dectin-1 in the patients is similar to human Dectin-1^∆CRD^ (1-114), which we have shown to retain ligand activity to CLEC-2 (original Figure 3C). Thus, the expression of truncated Dectin-1 retaining the stalk region may act as a ligand in the patients, which may contribute to the normal blood-lymph separation. Alternatively, we cannot exclude the possibility that, in human, Podoplanin may compensate for the impaired function of Dectin-1. We have described these points in the Discussion section of the revised manuscript (page 14, line 323).

5. The authors mainly use an NFAT-GFP reporter cell line as a proxy for hDectin-1 binding. When referring to this activity it would be preferable to state this more precisely rather than say "interacts with hDectin-1" examples on lines 93 and 97.

We apologize for our misleading description about the results. We have now changed this to the precise description in the Results section (page 4, line 97 and page 5, lines 101) of the revised manuscript.

6. A few of the graphs were difficult to read the x-axis e.g. Figure 1G – requires more space or remove some of the major ticks.

In accordance with the reviewer’s suggestion, we have corrected Figure 1G.

7. The SPR data in Figure 5F is not convincing and does not affect any of the conclusions, but this could be improved. The amount of binding signal for the positive (just 3RU) is very small. I appreciate you are using low molecular mass peptides for this but designing the experiment with a higher mass analyte would give better data. Why not immobilise the peptides and use the CLEC-2 CRD as the analyte? Or a longer peptide?

We apologize for not clearly describing the limitation of the study in Figure 5F. In accordance with the reviewer’s suggestion, we have newly synthesized a biotinylated 11-mer peptide with and without a disialylated core-1 *O*-glycan (see Author response image 3). Although these peptides were successfully immobilized onto a streptavidin sensor chip, we could not detect the apparent binding of CLEC-2 CRD even when using 150 μM (see Author response image 3). Furthermore, we also tested the CLEC-2 binding using amine-coupling immobilization, but we could not detect a binding response (see Author response image 3). We used 11-mer peptides to confirm the minimum moiety mediating the ligand function. As the reviewer suggested, we also synthesized longer glycopeptide, but we could not synthesize enough amounts of glycopeptide due to technical difficulties. In the present study, we performed multiple approaches to reach the current conclusion regarding the responsible region for the ligand activity. As the reviewer pointed out, Figure 5F itself cannot give us a strong conclusion. We therefore corrected our overstated interpretation in the Results section (page 9, line 211) in the revised manuscript.

**Author response image 3. sa2fig3:**